# Finding Interior Optimum of Black-box Constrained Objective with Bayesian Optimization

**Fengxue Zhang** *
University of Chicago

**Zejie Zhu**
University of Chicago

**Yuxin Chen**
University of Chicago

## Abstract

Optimizing objectives under constraints, where both the objectives and constraints are black box functions, is common in real-world applications such as medical therapy design, industrial process optimization, and hyperparameter optimization. Bayesian Optimization (BO) is a popular approach for tackling these complex scenarios. However, constrained Bayesian Optimization (CBO) often relies on heuristics, approximations, or relaxation of objectives, leading to weaker theoretical guarantees compared to canonical BO. In this paper, we address this gap by focusing on identifying the interior optimum of the constrained objective, deliberately excluding boundary candidates susceptible to noise perturbations. Our approach leverages the insight that jointly optimizing the objective and learning the constraints can help pinpoint high-confidence *regions of interest* (ROI) likely to contain the interior optimum. We introduce an efficient CBO framework, which intersects these ROIs within a discretized search space to determine a general ROI. Within this ROI, we optimize the acquisition functions, balancing the learning of the constraints and the optimization of the objective. We showcase the efficiency and robustness of our proposed CBO framework through the high probability regret bounds for the algorithm and extensive empirical validation.

## 1 Introduction

Bayesian optimization (BO) has been extensively studied over the past few decades as a powerful framework for addressing expensive black-box optimization tasks in machine learning, engineering, and science. In many real-world applications, these optimization tasks often involve black-box constraints that are costly to evaluate Digabel and Wild [2015]. Examples include choosing from a plethora of untested medical therapies under safety constraints [Sui et al., 2015]; determining optimal pumping rates in hydrology to minimize operational costs under constraints on plume boundaries [Gramacy et al., 2016]; or tuning hyperparameters of a neural network under memory constraints [Gelbart et al., 2014]. It is common to model constraints analogously to the objectives via Gaussian processes (GP) and then utilize an acquisition function to trade off the learning and optimization to decide subsequent query points.

Recently, significant advancements have been made in several directions to address constrained BO (CBO). For instance, extended Expected Improvement approaches [Bernardo et al., 2011, Gelbart et al., 2014, Gardner et al., 2014, Zhang et al., 2021, Bachoc et al., 2020] learn the constraints passively and calibrate the acquisition with feasibility. The augmented lagrangian (AL) methods [Gramacy et al., 2016, Picheny et al., 2016, Ariafar et al., 2019] convert constrained optimization into unconstrained optimization with additional hyperparameters. The entropy-based methods [Takeno et al., 2022] optimize the lower bound of the mutual information concerning the underlying optimum within the feasible region.

Workshop on Bayesian Decision-making and Uncertainty, 38th Conference on Neural Information Processing Systems (NeurIPS 2024).

---

*Corresponding to `zhangfx@uchicago.edu`

In this paper, we propose a novel framework that integrates active learning for level-set estimation (AL-LSE) [Gotovos et al., 2013, Nguyen et al., 2021] with BO for constrained Bayesian optimization. Our approach leverages the theoretical advantages of both paradigms, allowing for a rigorous performance analysis of the CBO method. A brief illustration of the framework design is shown in figure 2. The subsequent sections of this paper are structured as follows. First, we formally state the CBO problem and discuss the definition of a probabilistic regret as a performance metric that enables rigorous performance analysis. Following the problem statement, we propose the novel CBO framework, offer the corresponding performance analysis, and provide empirical evidence for the efficacy of the proposed algorithm. Finally, we reflect on the key takeaways of our framework and discuss its potential implications for future work.

## 2 Problem Statement

This section introduces a few useful notations and formalizes the problem. Consider a compact search space $\mathbf{X} \subseteq \mathbb{R}^d$. We aim to find a maximizer $\mathbf{x}^* \in \arg\max_{\mathbf{x} \in \mathbf{X}} f(\mathbf{x})$ of a black-box function $f : \mathbf{X} \to \mathbb{R}$, subject to $M$ black-box constraints $\mathcal{C}_m(\mathbf{x})$ ($m \in \mathbf{M} = \{1, 2, 3, ..., M\}$) such that each constraint is satisfied by staying above its corresponding threshold $h_m$.

For simplicity and without loss of generality, we let all $h_m = 0$. Thus, formally, our goal can be formulated as finding the *interior optimum*:

$$\max_{\mathbf{x} \in \mathbf{X}} f(\mathbf{x}) \text{ s.t. } \mathcal{C}_m(\mathbf{x}) > 0, \forall m \in \mathbf{M}$$

We maintain a Gaussian process ($\mathcal{GP}$) as the surrogate model for each black-box function, pick a point $\mathbf{x}_t \in \mathbf{X}$ at iteration $t$ by maximizing the acquisition function $\alpha : \mathbf{X} \to \mathbb{R}$, and observe the function values perturbed by additive noise: $y_{f,t} = f(\mathbf{x}_t) + \epsilon$ and $y_{\mathcal{C}_m,t} = \mathcal{C}_m(\mathbf{x}_t) + \epsilon$, with $\epsilon \sim \mathcal{N}(0, \sigma^2)$ being i.i.d. Gaussian noise.

The definition of reward plays an important role in analyzing online learning algorithms. Throughout the rest of the paper, we define the reward of CBO as the following and defer the detailed discussion of alternative reward choices to Appendix E.

$$r(\mathbf{x}_t) = \begin{cases} y_{f,t} & \text{if } \mathbb{I}(y_{\mathcal{C}_m(\mathbf{x}_t)} \geq 0) \quad \forall m \in \mathbf{M} \\ -\inf & \text{o.w.} \end{cases} \tag{1}$$

We want to locate the global maximizer efficiently

$$\mathbf{x}^* = \arg\max_{\mathbf{x} \in \mathbf{X}, \forall m \in \mathbf{M}, \mathcal{C}_m(\mathbf{x}) > 0} f(\mathbf{x})$$

## 3 The COBAR Algorithm

We start by introducing necessary notions from active learning for level-set estimation, followed by a detailed description of our proposed algorithm. The pseudocode and additional discussion are available in Appendix H.

### 3.1 Active learning for level-set estimation

We follow the common practice and assume the objective and each unknown constraint is sampled from a corresponding independent Gaussian process ($\mathcal{GP}$) [Hernández-Lobato et al., 2015, Gelbart et al., 2014, Gotovos et al., 2013] to treat the epistemic uncertainty. We could derive pointwise confidence interval estimation with the $\mathcal{GP}$ for each black-box function. We define the upper confidence bound $\text{UCB}_t(\mathbf{x}) \triangleq \mu_{t-1}(\mathbf{x}) + \beta_t^{1/2}\sigma_{t-1}(\mathbf{x})$ and lower confidence bound $\text{LCB}_t(\mathbf{x}) \triangleq \mu_{t-1}(\mathbf{x}) - \beta_t^{1/2}\sigma_{t-1}(\mathbf{x})$, where $\sigma_{t-1}(\mathbf{x}) = k_{t-1}(\mathbf{x}, \mathbf{x})^{1/2}$ and $\beta_t$ acts as a scaling factor corresponding to certain confidence.

For each unknown constraint $\mathcal{C}_m$, we follow the notations from Gotovos et al. [2013] and define the superlevel-set to be the areas that meet the constraint $\mathcal{C}_m$ with high confidence $S_{\mathcal{C}_m,t} \triangleq \{\mathbf{x} \in \mathbf{X} \mid \text{LCB}_{\mathcal{C}_m,t}(\mathbf{x}) > 0\}$. We define the sublevel-set to be the areas that do not meet the constraint $\mathcal{C}_m$ with high confidence $L_{\mathcal{C}_m,t} \triangleq \{\mathbf{x} \in \mathbf{X} \mid \text{UCB}_{\mathcal{C}_m,t}(\mathbf{x}) < 0\}$, and the undecided set is defined as $U_{\mathcal{C}_m,t} \triangleq \{\mathbf{x} \in \mathbf{X} \mid \text{UCB}_{\mathcal{C}_m,t}(\mathbf{x}) \geq 0, \text{LCB}_{\mathcal{C}_m,t}(\mathbf{x}) \leq 0\}$, where the points remain to be classified.

## 3.2 Region of interest identification for efficient CBO

In the CBO setting, we only care about the superlevel-set $S_{\mathcal{C}_m,t}$ and undecided-set $U_{\mathcal{C}_m,t}$, where the global optimum is likely to lie in. Hence, we define the region of interest for each constraint function $\mathcal{C}_m$ as $\hat{\mathbf{X}}_{\mathcal{C}_m,t} \triangleq S_{\mathcal{C}_m,t} \cup U_{\mathcal{C}_m,t} = \{\mathbf{x} \in \mathbf{X} \mid \mathrm{UCB}_{\mathcal{C}_m,t}(\mathbf{x}) \geq 0\}$. Similarly, for the objective function, though there is no pre-specified threshold, we could use the maximum of $\mathrm{LCB}_f(\mathbf{x})$ on the intersection of superlevel-set $S_{\mathcal{C},t} \triangleq \bigcap_m^M S_{\mathcal{C}_m,t}$

$$\mathrm{LCB}_{f,t,\max} \triangleq \begin{cases} \max_{\mathbf{x} \in S_{\mathcal{C},t}} \mathrm{LCB}_{f,t}(\mathbf{x}), & \text{if } S_{\mathcal{C},t} \neq \emptyset \\ -\infty, & \text{o.w.} \end{cases}$$

as the high confidence threshold for the $\mathrm{UCB}_{f,t}(\mathbf{x})$ to identify a region of interest for the optimization of the objective. Given that $\mathrm{UCB}_{f,t}(\mathbf{x}^*) \geq f^* \geq f(\mathbf{x}) \geq \mathrm{LCB}_{f,t}(\mathbf{x})$ with the probability specified by the choice of $\beta_t$, we define the ROI for the objective optimization as $\hat{\mathbf{X}}_{f,t} \triangleq \{\mathbf{x} \in \mathbf{X} \mid \mathrm{UCB}_{f,t}(\mathbf{x}) \geq \mathrm{LCB}_{f,t,\max}\}$. By taking the intersection of the ROI of each constraint, we could identify the ROI for identifying the feasible region $\hat{\mathbf{X}}_{\mathcal{C},t} \triangleq \bigcap_m^M \hat{\mathbf{X}}_{\mathcal{C}_m,t}$. The combined ROI for CBO is determined by intersecting the ROIs of constraints and the objective:

$$\hat{\mathbf{X}}_t \triangleq \hat{\mathbf{X}}_{f,t} \cap \hat{\mathbf{X}}_{\mathcal{C},t} \tag{2}$$

## 3.3 Combining acquisition functions for CBO

**Acquisition function for optimizing the objective** To optimize the unknown objective $f$ when $\hat{\mathbf{X}}_t$ is established, we can employ the following acquisition function [2]

$$\alpha_{f,t}(\mathbf{x}) \triangleq \begin{cases} \mathrm{UCB}_{f,t}(\mathbf{x}) - \mathrm{LCB}_{f,t,\max} & S_{\mathcal{C},t} \neq \emptyset \\ \mathrm{UCB}_{f,t}(\mathbf{x}) - \mathrm{LCB}_{f,t}(\mathbf{x}) & \text{o.w.} \end{cases} \tag{3}$$

At given $t$, to efficiently optimize the black-box $f$ we evaluate the point $\mathbf{x}_t = \arg\max_{\mathbf{x} \in \hat{\mathbf{X}}_t} \alpha_{f,t}(\mathbf{x})$. Since at a given $t$, when $\mathrm{LCB}_{f,t,\max}$ is constant, the acquisition function is equivalent to $\mathrm{UCB}_{f,t}(\mathbf{x})$.

**Acquisition function for learning the constraints** When we merely focus on identifying the feasible region defined by a certain unknown constraint $\mathcal{C}_k$, we could apply the following active learning acquisition function.

$$\alpha_{\mathcal{C}_m,t}(\mathbf{x}) \triangleq \mathrm{UCB}_{\mathcal{C}_m,t}(\mathbf{x}) - \mathrm{LCB}_{\mathcal{C}_m,t}(\mathbf{x}) \tag{4}$$

At given $t$, we evaluate the point $\mathbf{x}_t = \arg\max_{\mathbf{x} \in U_{\mathcal{C}_m,t} \cap \hat{\mathbf{X}}_t} \alpha_{\mathcal{C}_m,t}(\mathbf{x})$ to efficiently identify the feasible region defined by $\mathcal{C}_m$. Note that the acquisition function $\alpha_{\mathcal{C}_m,t}(\mathbf{x})$ is not maximized on the full $\hat{\mathbf{X}}_{\mathcal{C}_m,t}$, but only on $U_{\mathcal{C}_m,t} \cap \hat{\mathbf{X}}_t$. The active learning on the superlevel-set $S_{\mathcal{C}_m,t} \cap \hat{\mathbf{X}}_t$ doesn't contribute to identifying the corresponding feasible region.

**The COBAR acquisition criterion** With the two acquisitions discussed above and the ROIs discussed in section 3.2, we propose the algorithm **CO**nstrained **B**O through **A**daptive **R**egion of Interest Acquisition (COBAR). COBAR essentially picks a data point with the maximum acquisition function value across all the acquisition functions defined on different domains. The maximization of different acquisition functions allows an adaptive tradeoff between the active learning of the constraints and the Bayesian Optimization of the objective on the feasible region. The intersection of ROIs allows for efficient search space shrinking for CBO. The full procedure is detailed in algorithm 2. We also illustrate the detailed procedure on a 1D toy example in figure 3 of Appendix A.

## 3.4 Theoretical Analysis

In the following, we show that we could bound the simple regret of COBAR after sufficient rounds. Concretely, in Theorem 1, we provide an upper bound on the width of the confidence interval for the global optimum $f^* = f(\mathbf{x}^*)$. We defer additional results and proofs to Appendix B[3].

---

[2]Same criterion has been studied under the unconstrained setting [Zhang et al., 2023].

[3]With additional assumptions on the regularization of the underlying function, we derive the analogous analysis on continuous search space in Appendix C.

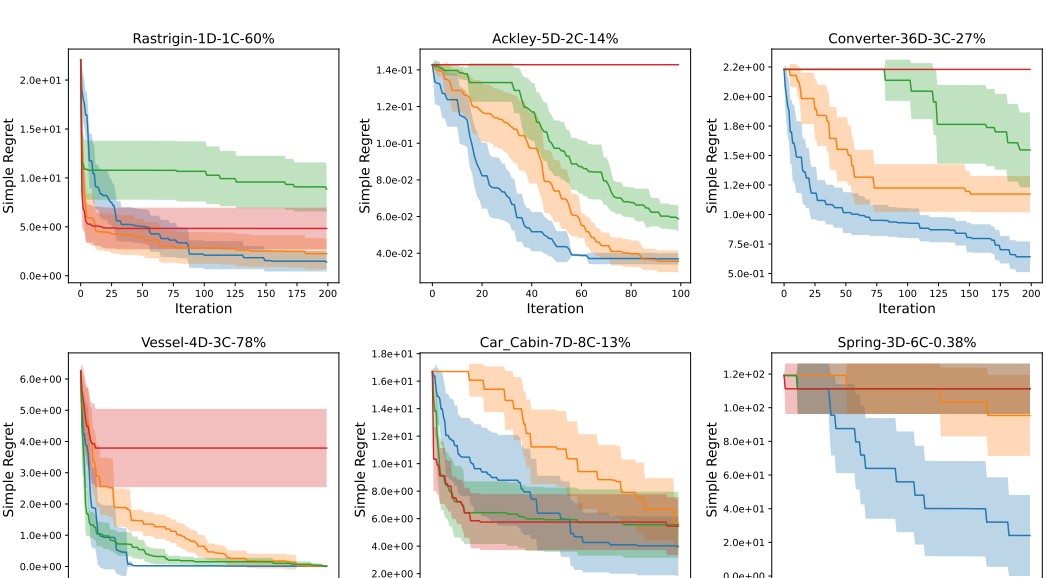

Figure 1: The input dimensionality, the number of constraints, and the approximate portion of the feasible region in the whole search space for each task are denoted on the titles. We run the algorithms on each task for at least 15 independent trials. The curves show the average simple regret after standardization, while the shaded area denotes the 95% confidence interval through the optimization.

**Theorem 1.** *Under the proper assumptions, with a constant* $\beta = 2\log(\frac{2(M+1)|\tilde{D}|T}{\delta})$ *and the acquisition function from* $algorithm$ 2, *there exists an* $\epsilon_f \leq \epsilon_C$, *such that after at most* $T \geq \frac{\beta\widehat{\gamma_T}C_1}{\epsilon_f^2}$ *iterations, we have* $\mathbb{P}\left[|CI_{f^*,T}| \leq \epsilon_f, f^* \in CI_{f^*,T}\right] \geq 1 - \delta$ *Here,* $C_1 = 8/\log(1 + \sigma^{-2})$.

## 4 Experiments

We empirically study the performance of COBAR against three baselines, including (1) cEI, the extension of EI into CBO from Gelbart et al. [2014], (2) cMES-IBO, a state-of-the-art information-based approach by Takeno et al. [2022], and (3) SCBO, a recent Thompson Sampling (TS) method tailored for scalable CBO from Eriksson and Poloczek [2021]. The results are illustrated in figure 1. We abstain from comparison against Augmented-Lagrangian methods, following the practice of Takeno et al. [2022], as past studies have illustrated its inferior performance against sampling methods [Eriksson and Poloczek, 2021] or information-based methods [Takeno et al., 2022, Hernández-Lobato et al., 2014]. We defer the comparison against CONFIG Xu et al. [2023] to Appendix G, due to the difference in objective and a resulting instability on our benchmarks. We compare COBAR against the aforementioned baselines across six CBO tasks. The first two synthetic CBO tasks are constructed from conventional BO benchmark tasks [Balandat et al., 2020]. Among the other four real-world CBO tasks, the first three are extracted from Tanabe and Ishibuchi [2020], offering a broad selection of multi-objective multi-constraints optimization tasks. The fourth one is a 32-dimensional optimization task extracted from the UCI Machine Learning repository [mis, 2019]. Further details about the datasets are available in Appendix F.

## 5 Conclusion

Bayesian optimization with unknown constraints poses challenges in the adaptive tradeoff between optimizing the unknown objective and learning the constraints. We introduce COBAR, which is backed by rigorous theoretical guarantees, to efficiently address constrained Bayesian optimization. Our key insights include: (1) the ROIs determined through adaptive level-set estimation can congregate and contribute to the overall Bayesian optimization task; (2) acquisition functions based on independent GPs can be unified in a principled way. Through extensive experiments, we validate the efficacy and robustness of our proposed method across various tasks finding the interior optimum.

## Acknowledgment

This work was supported in part by the National Science Foundation under Grant No. IIS 2313131, IIS 2332475, and CMMI 2037026. The authors acknowledge the University of Chicago's Research Computing Center for their support of this work.

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

# A   Additional Illustration

We put the illustration of the pipeline of proposed algorithm COBAR here.

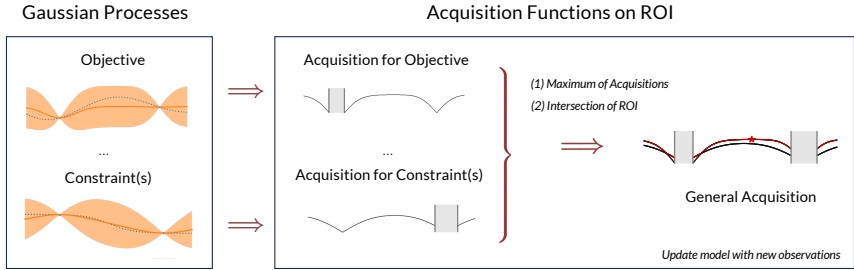

Figure 2: Pipeline of proposed algorithm COBAR. In the left box, we maintain a Gaussian process as the surrogate model for the unknown objective and each constraint. The dotted curve shows the actual function, the red curve shows the predicted mean, and the shaded area denotes the confidence interval. In the right box, we first derive the acquisitions from each Gaussian process defined on a corresponding region of interests and define the general acquisition function by combining them all. Each time, the algorithm maintains the model, maximizes the general acquisition function to pick the candidate to evaluate, and then updates the model with the new observation. In the Algorithm section, we will elaborate on the filtered gray gap in the acquisition.

We put the illustration of a 1D Toy Example in figure 3.

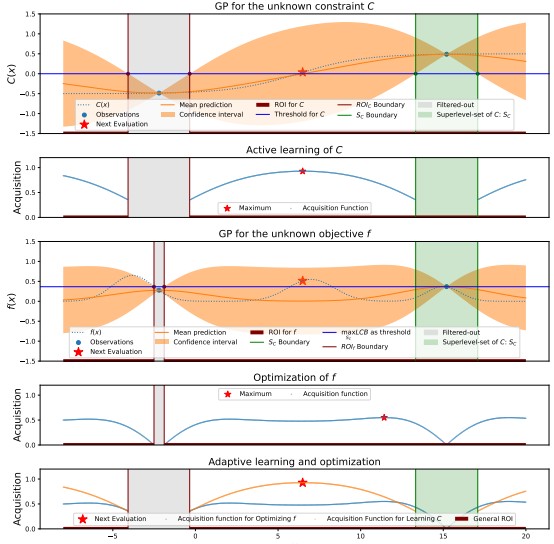

Figure 3: Illustration of COBAR on a synthetic noise-free 1D example. The first two rows show the GP for the $\mathcal{C}$, the superlevel-set $S_{\mathcal{C}}$, the region of interest $\hat{\mathbf{X}}_{\mathcal{C}}$ and the corresponding acquisition function $\alpha_{\mathcal{C}_m,t}(\mathbf{x})$ as defined in equation 4. The following two rows show the GP for $f$, the region of interest $\hat{\mathbf{X}}_f$, and the corresponding acquisition function $\alpha_{f,t}(\mathbf{x})$ defined in equation 3. We show that after identifying $S_{\mathcal{C}}$, we could define the threshold for ROI identification of $f$ accordingly. The bottom row demonstrates that the general ROI $\hat{\mathbf{X}}$ as defined in equation 2 is identified by taking the intersection ROI for $f$ and $\mathcal{C}$. The general acquisition function is defined as the maximum of the acquisition for $f$ and $\mathcal{C}$ and is maximized on the $\hat{\mathbf{X}}$. The scaling and length scale of the GPs are learned via maximum likelihood estimation.

## B Analysis and Missing Proofs

### B.1 Assumptions

We first state a few assumptions that provide insights into the convergence properties of COBAR. The first one follows Srinivas et al. [2009] as a standard assumption for BO.

**Assumption 1.** *The objective and constraints are sampled from independent Gaussian processes. Formally, for all $t < T$ and $\mathbf{x} \in \mathbf{X}$, $f(\mathbf{x})$ is a sample from $\mathcal{GP}_{f,t}$, and $\mathcal{C}_m(\mathbf{x})$ is a sample from $\mathcal{GP}_{\mathcal{C}_m,t}$, for all $m \in \mathbf{M}$.*

**Assumption 2.** *A global optimum exists within the feasible region. The distance between this global optimum and the boundaries of the feasible regions is uniformly bounded below by $\epsilon_{\mathcal{C}}$. More specifically, for all $m \in \mathbf{M}$, $\exists \epsilon_m > 0$ such that $\mathcal{C}_m(\mathbf{x}^*) > \epsilon_m$, then it holds that $\mathcal{C}_m(\mathbf{x}^*) > \epsilon_{\mathcal{C}} = \min_{m \in \mathbf{M}} \epsilon_m$.*

**Assumption 3.** *Given a proper choice of $\beta_t$ that is non-increasing, the confidence intervals are consistent. Concretely, $\forall t_1 < t_2 < T$ and $\mathbf{x} \in \mathbf{X}$, if $\beta_{t_1} \geq \beta_{t_2}$, then $UCB_{t_1}(\mathbf{x}) \geq UCB_{t_2}(\mathbf{x})$ and $LCB_{t_1}(\mathbf{x}) \leq LCB_{t_2}(\mathbf{x})$.*

This is a mild assumption as long as $\beta_t$ is non-increasing, given recent work by Koepernik and Pfaff [2021] showing that if the kernel is continuous and the sequence of sampling points lies sufficiently dense, the variance of the posterior $\mathcal{GP}$ converges to zero almost surely monotonically if the function is in metric space. If the assumption is violated, the technique of taking the intersection of all historical confidence intervals introduced by Gotovos et al. [2013] could similarly guarantee a monotonically shrinking confidence interval. That is, when $\exists t_1 < t_2 < T, \mathbf{x} \in \mathbf{X}$, if we have $\text{UCB}_{t_1}(\mathbf{x}) < \text{UCB}_{t_2}(\mathbf{x})$ or $\text{LCB}_{t_1}(\mathbf{x}) > \text{LCB}_{t_2}(\mathbf{x})$, we let $\text{UCB}_{t_2}(\mathbf{x}) = \text{UCB}_{t_1}(\mathbf{x})$ or $\text{LCB}_{t_2}(\mathbf{x}) = \text{LCB}_{t_1}(\mathbf{x})$ to guarantee the monotonocity. To allow for a plug-in of the intersection technique, and without loss of accuracy, we keep using the notation UCB and LCB without further parsing the value in the following discussion of algorithm design and theoretical analysis. The cost of violating the Assumption 3 has been studied in corollary 3 by Zhang et al. [2023]. We refrain from repeating the analysis here.

### B.2 Lemma 1

The following lemma justifies the definition of the regions(s) of interest $\hat{\mathbf{X}}_t$ defined in equation 2. For clarity, we denote $\tilde{D}_{\hat{\mathbf{X}}_t} = \tilde{D} \cap \hat{\mathbf{X}}_t$, and $CI_{f^*,t} = [\max_{\mathbf{x} \in \tilde{D}_{\hat{\mathbf{X}}_t}} \text{LCB}_t(\mathbf{x}), \max_{\mathbf{x} \in \tilde{D}_{\hat{\mathbf{X}}_t}} \text{UCB}_t(\mathbf{x})]$.

**Lemma 1.** *Under the assumptions above, the regions of interest $\hat{\mathbf{X}}_t$, as defined in equation 2, contain the global optimum with high probability. Formally, for all $\delta \in (0, 1)$, $T \geq t \geq 1$, and any finite discretization $\tilde{D}$ of $\mathbf{X}$ that contains the optimum $\mathbf{x}^* = \arg\max_{\mathbf{x} \in \mathbf{X}} f(\mathbf{x})$ where $\mathcal{C}_m(\mathbf{x}^*) > \epsilon_{\mathcal{C}}$ for all $m \in \mathbf{M}$ and $\beta_t = 2\log(2(M+1)|\tilde{D}|\pi_t/\delta)$ with $\sum_{T \geq t \geq 1} \pi_t^{-1} = 1$, we have $\mathbb{P}\left[\mathbf{x}^* \in \tilde{D}_{\hat{\mathbf{X}}_t}\right] \geq 1 - \delta$.*

To guarantee $\beta_t$ to be non-increasing, we could let $\pi_t = T$ and therefore $\beta = 2\log(\frac{2(M+1)|\tilde{D}|T}{\delta})$ is a constant. The lemma shows that with proper choice of prior and $\beta$, the $\hat{\mathbf{X}}_{f,t}$ remains nonempty during optimization.

Subsequently, let's define the maximum information gain about function $f$ after $T$ rounds: $\gamma_{f,T} = \max_{A \subset \tilde{D}:|A|=T} \mathbb{I}(y_A; f_A)$ and

$$\widehat{\gamma_T} = \sum_{g \in \{f\} \cup \{\mathcal{C}_m\}_{m \in \mathbf{M}}} \gamma_{g,T} \tag{5}$$

*Proof.* With probability at least $1 - 1/2\delta$, $\forall \mathbf{x} \in \tilde{D}, \forall T \geq t \geq 1, \forall g \in \{f\} \cup \{\mathcal{C}_m\}_{m \in \mathbf{M}}$,

$$|g(\mathbf{x}) - \mu_{g,t-1}(\mathbf{x})| \leq \beta_t^{1/2} \sigma_{g,t-1}(\mathbf{x})$$

Note that we also take the union bound on $g \in \{f\} \cup \{\mathcal{C}_m\}_{m \in \mathbf{M}}$.

This is similarly derived as lemma 5.1 of Srinivas et al. [2009] or lemma 1 of Zhang et al. [2023]. *Different from previous proofs*, we do not require the lemma to hold for $\forall t \geq 1$. Instead, we require it

to hold for $\forall T \geq t \geq 1$. This alleviates the need of the convergence of the series $\sum_{t \geq 1} \pi_t^{-1} = 1$ to $\sum_{t \geq 1}^{T} \pi_t^{-1} = 1$ when taking the union bound. Specifically, we could set $\pi_t = T$, which essentially makes $\beta_t = 2 \log(\frac{2(M+1)|\tilde{D}|T}{\delta})$ a constant. Hence, we use the $\beta$ in the following instead of $\beta_t$ as traditionally used to highlight this difference.

First, by definition $S_{\mathcal{C},t} \triangleq \bigcap_m^{\mathbf{M}} S_{\mathcal{C}_m,t}$, we have $\forall t \leq T, \mathbf{x} \in \tilde{D} \cap S_{\mathcal{C},t}, \forall m \in \mathbf{M}$

$$\mathbb{P}\left[\mathcal{C}_m(\mathbf{x}) \geq \text{LCB}_{\mathcal{C}_m,t}(\mathbf{x}) > 0\right] \geq 1 - 1/2\delta$$

meaning with probability at $1 - \delta$, $\mathbf{x}$ lies in the feasible region. At the same time, we have, $\forall t \leq T$, $\forall m \in \mathbf{M}$, given $\mathcal{C}_m(\mathbf{x}) > 0$

$$\mathbb{P}\left[\text{UCB}_{f,t}(\mathbf{x}^*) \geq f(\mathbf{x}^*) \geq f(\mathbf{x}) \geq \text{LCB}_{f,t}(\mathbf{x})\right] \geq 1 - 1/2\delta$$

Given the mutual independency between the objective $f$ and the constraints $\mathcal{C}_m$, and by the definition of the threshold $\text{LCB}_{f,t,\max}$, we have $\forall t \leq T$, when $\exists \mathbf{x} \in \tilde{D} \cap S_{\mathcal{C},t}$,

$$\mathbb{P}\left[\text{UCB}_{f,t}(\mathbf{x}^*) > \text{LCB}_{f,t,\max}\right] \geq 1 - \delta$$

Note when $\tilde{D} \cap S_{\mathcal{C},t} = \emptyset$, $\text{LCB}_{f,t,\max} = -\infty$, we have $\mathbb{P}\left[\text{UCB}_{f,t}(\mathbf{x}^*) > \text{LCB}_{f,t,\max}\right] = 1$.

In summary, we've shown that with probability at least $1 - \delta$, $\mathbf{x}^* \in \tilde{D} \cap \hat{\mathbf{X}}_{f,t}$.

Next, by the definition of $\mathbf{x}^* = \arg\max_{\mathbf{x} \in \mathbf{X}} f(\mathbf{x})$ $s.t.$ $\mathcal{C}_m(\mathbf{x}^*) > \epsilon_{\mathcal{C}}$ we have $\forall t \leq T, \forall m \in \mathbf{M}$

$$\mathbb{P}\left[\text{UCB}_{\mathcal{C}_m,t}(\mathbf{x}^*) \geq \mathcal{C}_m(\mathbf{x}^*) > 0\right] \geq 1 - 1/2\delta$$

meaning with probability at least $1 - 1/2\delta$, $\mathbf{x}^* \in \tilde{D} \cap \hat{\mathbf{X}}_{\mathcal{C}_m,t}$. And in general, we have $\forall t \leq T, \forall m \in \mathbf{M}$

$$\mathbb{P}\left[\mathbf{x}^* \in \tilde{D} \cap \hat{\mathbf{X}}_t\right] \geq 1 - \delta$$

$\square$

### B.3 Proof of Theorem 1

The following lemmas show that the maximum of the acquisition functions equation 3 and 4 are both bounded after sufficient evaluations.

**Lemma 2.** *Under the conditions assumed in Theorem 1 except for Assumption 2, let $\alpha_t = \max_{g \in \mathcal{G}} \alpha_{g,t}(\mathbf{x}_{g,t})$ as in Algorithm 2, with $\beta = 2 \log(\frac{2(M+1)|\tilde{D}|T}{\delta})$ that is a constant, after at most $T \geq \frac{\beta \widehat{\gamma_T} C_1}{\epsilon_f^2}$ iterations, $\alpha_T \leq \epsilon_f$ Here $C_1 = 8/\log(1 + \sigma^{-2})$.*

The inequation $T \geq \frac{\beta \widehat{\gamma_T} C_1}{\epsilon_f^2}$ has $T$ on both side, which follows the convention in Gotovos et al. [2013].

*Proof.* We first unify the notation in the acquisition functions.
$\forall T \geq t \geq 1, \forall g \in \{\mathcal{C}_m\}_{m \in \mathbf{M}}$, when $\tilde{D}_{\hat{\mathbf{X}}_t} \cap U_{g,t} \neq \emptyset$,

$$\max_{\mathbf{x} \in \tilde{D}_{\hat{\mathbf{X}}_t} \cap U_{g,t}} \text{UCB}_{g,t}(\mathbf{x}) - \text{LCB}_{g,t}(\mathbf{x}) \leq \alpha_t \tag{6}$$

$\forall T \geq t \geq 1, \forall g \in \{\mathcal{C}_m\}_{m \in \mathbf{M}}$, when $\tilde{D}_{\hat{\mathbf{X}}_t} \cap U_{\mathcal{C}_m,t} = \emptyset$, let

$$\max_{\mathbf{x} \in \tilde{D}_{\hat{\mathbf{X}}_t} \cap U_{g,t}} \text{UCB}_{g,t}(\mathbf{x}) - \text{LCB}_{g,t}(\mathbf{x}) = 0 \leq \alpha_t \tag{7}$$

$\forall T \geq t \geq 1, g = f$, when $S_{\mathcal{C},t} = \emptyset$, we have

$$\max_{\mathbf{x} \in \tilde{D}_{\hat{\mathbf{X}}_t}} \text{UCB}_{f,t}(\mathbf{x}) - \text{LCB}_{f,t}(\mathbf{x}) \leq \alpha_t \tag{8}$$

$\forall T \geq t \geq 1, g = f$, when $S_{\mathcal{C},t} \neq \emptyset$, we have

$$\max_{\mathbf{x} \in \tilde{D}_{\hat{\mathbf{X}}_t}} \text{UCB}_{f,t}(\mathbf{x}) - \text{LCB}_{f,t.max} \leq \alpha_t \tag{9}$$

By lemma 5.1, 5.2 and 5.4 of Srinivas et al. [2009], with $\beta = 2\log(\frac{2(M+1)|\tilde{D}|T}{\delta})$, $\forall g \in \{f\} \cup \{\mathcal{C}_m\}_{m\in\mathbf{M}}$ and $\forall x_t \in \tilde{D}_{\hat{\mathbf{X}}_t} \subseteq \tilde{D}$, we have $\sum_{t=1}^{T}(2\beta^{1/2}\sigma_{g,t-1},(\mathbf{x}_t))^2 \leq C_1\beta\gamma_{g,T}$. By definition of $\alpha_t$ , we have the following

$$\sum_{t=1}^{T}\alpha_t^2 \leq \sum_{t=1}^{T}\max_{g\in\{f\}\cup\{\mathcal{C}_m\}_{m\in\mathbf{M}}}(2\beta^{1/2}\sigma_{g,t-1}(\mathbf{x}_{g,t}))^2$$

$$\leq \sum_{t=1}^{T}\sum_{g\in\{f\}\cup\{\mathcal{C}_m\}_{m\in\mathbf{M}}}(2\beta^{1/2}\sigma_{g,t-1}(\mathbf{x}_t))^2$$

$$\leq \sum_{g\in\{f\}\cup\{\mathcal{C}_m\}_{m\in\mathbf{M}}}C_1\beta\gamma_{g,T}$$

$$= C_1\beta\widehat{\gamma_T}$$

The last line holds due to the definition in equation 5. By Cauchy-Schwarz, we have

$$\frac{1}{T}(\sum_{t=1}^{T}\alpha_t)^2 \leq C_1\beta\widehat{\gamma_T}$$

With $Assumption$ $3$, $\forall g \in \{\mathcal{C}_m\}_{m\in\mathbf{M}}, \forall 1 \leq t_1 < t_2 \leq T, \forall g \in \{\mathcal{C}_m\}_{m\in\mathbf{M}}$, we have $U_{g,t_2} \subseteq U_{g,t_1}$ and $\hat{\mathbf{X}}_{t_2} \subseteq \hat{\mathbf{X}}_{t_1}$, and most importantly, $\alpha_{t_2} \leq \alpha_{t_1}$. Therefore

$$\alpha_T \leq \frac{1}{T}\sum_{t=1}^{T}\alpha_t \leq \sqrt{\frac{C_1\beta\widehat{\gamma_T}}{T}}$$

As a result, after at most $T \geq \frac{\beta\widehat{\gamma_T}C_1}{\epsilon_f^2}$ iterations, we have $\alpha_T \leq \epsilon_f$. $\qquad\square$

With Lemma 2, we could first prove that after adequately $T$ rounds of evaluations such that $\epsilon_f \leq \min_{m\in\mathbf{M}}\epsilon_m$ is sufficiently small, with certain probability, $\mathbf{x}^* \in S_{\mathcal{C},T}$. Then $\mathrm{LCB}_{f,t,\max} \neq -\infty$, and therefore the width of $[\max_{\mathbf{x}\in\tilde{D}_{\hat{\mathbf{X}}_t}}\mathrm{LCB}_{f,T}(\mathbf{x}), \max_{\mathbf{x}\in\tilde{D}_{\hat{\mathbf{X}}_t}}\mathrm{UCB}_{f,T}(\mathbf{x})]$, which is a the high confidence interval of $f^*$, is bounded by $\epsilon_f$.

*Proof.* We first prove that after at most $T \geq \frac{\beta\widehat{\gamma_T}C_1}{\epsilon_f^2}$ iterations, $\mathbb{P}\left[\mathbf{x}^* \in \tilde{D}_{\hat{\mathbf{X}}_t} \cap S_{\mathcal{C},T}\right] \geq 1 - 1/2\delta$. Given equation 6 and 7 and Lemma 2, we have $\forall g \in \{\mathcal{C}_m\}_{m\in\mathbf{M}}$,

$$\max_{\mathbf{x}\in\tilde{D}_{\hat{X}_T}\cap U_{g,T}}\mathrm{UCB}_{g,T}(\mathbf{x}) - \mathrm{LCB}_{g,T}(\mathbf{x}) \leq \epsilon_f \leq \min_{m\in\mathbf{M}}\epsilon_m$$

According to the definition of $U_{g,T}$, $\forall \mathbf{x} \in \tilde{D}_{\hat{X}_T} \cap U_{g,T}$, $\forall g \in \{\mathcal{C}_m\}_{m\in\mathbf{M}}$

$$\mathrm{UCB}_{g,T}(\mathbf{x}) \leq \epsilon_f + \mathrm{LCB}_{g,T}(\mathbf{x}) \leq \epsilon_f \leq \min_{m\in\mathbf{M}}\epsilon_m$$

According to Assumption 2, and Lemma 1, $\forall m \in \mathbf{M}$, we have

$$\mathbb{P}\left[\mathrm{UCB}_{\mathcal{C}_m,T}(\mathbf{x}^*) > \max_{\mathbf{x}\in\tilde{D}_{\hat{\mathbf{X}}_t}\cap U_{\mathcal{C}_m,t}}\mathrm{UCB}_{\mathcal{C}_m,T}(\mathbf{x})\right] \geq 1 - 1/2\delta$$

Given $= \tilde{D}_{\hat{X}_T} \cap S_{\mathcal{C},T} = \tilde{D}_{\hat{\mathbf{X}}_t} \cap \hat{\mathbf{X}}_{\mathcal{C},T} \backslash \cup_{m\in\mathbf{M}} U_{\mathcal{C}_m,T}$, when $t = T$, we have

$$\mathbb{P}\left[\mathbf{x}^* \in \tilde{D}_{\hat{X}_T} \cap S_{\mathcal{C},T}\right] \geq 1 - 1/2\delta \qquad (10)$$

As a result

$$\mathbb{P}\left[\mathrm{LCB}_{f,T,\max} \neq -\infty\right] \geq 1 - 1/2\delta$$

Next, we prove the upper bound for the width of the high-confidence interval of $f^*$. Given that $\text{LCB}_{f,T,\max} \neq -\infty$, we have

$$\max_{\mathbf{x} \in \tilde{D}_{\hat{X}_T}} \text{UCB}_{f,T}(\mathbf{x}) - \max_{\mathbf{x} \in \tilde{D}_{\hat{X}_T}} \text{LCB}_{f,T}(\mathbf{x})$$

$$\leq \max_{\mathbf{x} \in \tilde{D}_{\hat{X}_T}} \text{UCB}_{f,T}(\mathbf{x}) - \text{LCB}_{f,T,\max}$$

$$\leq \alpha_T$$

$$\leq \epsilon_f$$

Combining it with the observation that with probability $1 - 1/2\delta$,

$$\max_{\mathbf{x} \in \tilde{D}_{\hat{X}_T}} \text{LCB}_{f,T}(\mathbf{x}) < f(\mathbf{x}^*) \leq \max_{\mathbf{x} \in \tilde{D}_{\hat{X}_T}} \text{UCB}_{f,T}(\mathbf{x})$$

we attain the final result that after $T \geq \frac{\beta \widehat{\gamma_T} C_1}{\epsilon^2}$ iterations,

$$\mathbb{P}\left[|CI_{f^*,T}| \leq \epsilon, f^* \in CI_{f^*,T}\right] \geq 1 - \delta$$

$\square$

## B.4 Corollary 1

One direct result of Theorem 1 is that if any point belongs to $\tilde{D}$ that lies in the feasible set defined by the unknown constraints bears a suboptimal gap on the reward except for the global optimum, then after sufficient query, the algorithm will identify $\mathbf{x}^*$ as the only point in the ROI. In that case, COBAR will only query $\mathbf{x}^*$ and achieve zero regret afterward.

**Corollary 1.** *We assume the aforementioned conditions hold, and $\forall \mathbf{x} \in \tilde{D}$, when $\forall m \in \mathbf{M}$, $\mathcal{C}_m(\mathbf{x}) > 0$, $\mathbf{x} \neq \mathbf{x}^*$, it holds that $\exists \epsilon_{\mathcal{C}} \geq 2\epsilon_f > 0$, $f^* - f(\mathbf{x}) > 2\epsilon_f$. In addition, we use $\beta = 2\log(\frac{2(M+1)|\tilde{D}|T}{\delta})$ and the acquisition function from Algorithm 2. After at most $t \geq \frac{\beta \widehat{\gamma_t} C_1}{\epsilon_f^2}$ iterations, we have $\mathbb{P}\left[\mathbf{R}_t = 0\right] \geq 1 - \delta$. Here, $C_1 = 8/\log(1 + \sigma^{-2})$ and $t \leq T$.*

*Proof.* We simply need to show that after $t \geq \frac{\beta \widehat{\gamma_t} C_1}{\epsilon_f^2}$ iterations, with probability at least $1 - \delta$, $\mathbf{x}^*$ is the only member in $\tilde{D}_{\hat{\mathbf{X}}_t}$.

Similar to Theorem 1, we have $\mathbb{P}\left[|CI_{f^*,t}| \leq \epsilon_f, f^* \in CI_{f^*,t}\right] \geq 1 - \delta$. At the same time, given the proof of Lemma 2, we have $\forall \mathbf{x} \in \tilde{D}_{\hat{\mathbf{X}}_t}$, $2\beta^{1/2}\sigma_{f,t-1}(\mathbf{x}) \leq \epsilon_f$.

Then if $\exists \mathbf{x} \neq \mathbf{x}^*$ and $\mathbf{x} \in \tilde{D}_{\hat{\mathbf{X}}_t}$, we have $f^* - f(\mathbf{x}) > 2\epsilon_f$, while

$$\mathbb{P}\left[f^* - f(\mathbf{x}) \leq |CI_{f^*,t}| + \text{UCB}_{f,t}(\mathbf{x}) - \text{LCB}_{f,t}(\mathbf{x}) \leq 2\beta^{1/2}\sigma_{f,t-1}(\mathbf{x}) + \epsilon_f \leq 2\epsilon_f\right] \geq 1 - \delta$$

This contradiction means with probability at least $1 - \delta$, $\mathbf{x}^*$ is the only member in $\tilde{D}_{\hat{\mathbf{X}}_t}$, and $\mathbf{x}_t = \mathbf{x}^*$. As a result, $\mathbb{P}\left[\mathbf{R}_t = 0\right] \geq 1 - \delta$, when $T \geq t \geq \frac{\beta \widehat{\gamma_t} C_1}{\epsilon_f^2}$. $\square$

## B.5 Corollary 2

Similarly, if a group of suboptimal candidates lies in the feasible area and is sufficiently close to $\mathbf{x}^*$, then Assumption 2 also holds for those suboptimal points. In this condition, the algorithm achieves a sublinear cumulative regret after identifying this near-optimal region.

**Corollary 2.** *We assume the aforementioned conditions hold, and $\forall \mathbf{x} \in \tilde{D}$, when $\forall m \in \mathbf{M}$, $\mathcal{C}_m(\mathbf{x}) > 0$, $\mathbf{x} \neq \mathbf{x}^*$, $\exists \epsilon_{\mathcal{C}} \geq \epsilon_f > 0$, $f^* - f(\mathbf{x}) \leq 2\epsilon_f$, it holds that $\forall m \in \mathbf{M}$, $\mathcal{C}_m(\mathbf{x}) \geq \epsilon_{\mathcal{C}}$. In addition, we use $\beta = 2\log(\frac{2(M+1)|\tilde{D}|T}{\delta})$ and the acquisition function from Algorithm 2. After at most $t' \geq \frac{\beta \widehat{\gamma_{t'}} C_1}{\epsilon_f^2}$ iterations, we have, $\mathbb{P}\left[\sum_{t=t'}^{T} r(\mathbf{x}^*) - r(\mathbf{x}_t) \leq \sqrt{(T-t')\beta \gamma_T C_1}\right] \geq 1 - \delta$. Here, $C_1 = 8/\log(1 + \sigma^{-2})$ and $t' \leq T$.*

*Proof.* We follow the same path as the proof of Corollary 1.

Similar to Theorem 1, we have $\mathbb{P}\left[|CI_{f^*,t}| \leq \alpha_t \leq \epsilon_f, f^* \in CI_{f^*,t}\right] \geq 1-\delta$. At the same time, given the proof of Lemma 2, we have $\forall \mathbf{x} \in \tilde{D}_{\hat{\mathbf{X}}_t}, 2\beta^{1/2}\sigma_{f,t-1}(\mathbf{x}) \leq \alpha_t \leq \epsilon_f$.

Then $\forall \mathbf{x} \neq \mathbf{x}^*$ and $\mathbf{x} \in \tilde{D}_{\hat{\mathbf{X}}_t}$, we have

$$\mathbb{P}\left[f^* - f(\mathbf{x}) \leq |CI_{f^*,t}| + \mathrm{UCB}_{f,t}(\mathbf{x}) - \mathrm{LCB}_{f,t}(\mathbf{x}) \leq 2\alpha_t \leq 2\epsilon_f\right] \geq 1-\delta$$

Then by assumption, $\forall \mathbf{x} \in \tilde{D}_{\hat{\mathbf{X}}_t}, \forall m \in \mathbf{M}$, we have probability at least $1-\delta$, $\mathcal{C}_m(\mathbf{x}) \geq \epsilon_\mathcal{C}$, and hence $\mathbf{x} \notin U_{\mathcal{C}_m,t}$. According to the algorithm, it regresses to GP-UCB by Srinivas et al. [2009] between $t'$ and $T$.

$$\sum_{t=t'}^{T}(r(\mathbf{x}^*) - r(\mathbf{x}_t))^2 \leq \beta C_1(\gamma_T - \gamma_{t'})$$

$$\leq \beta C_1 \gamma_T (1 - t'/T)$$

By Cauchy-Schwarz, we have

$$\sum_{t=t'}^{T}(r(\mathbf{x}^*) - r(\mathbf{x}_t)) \leq \sqrt{(T - t')\sum_{t=t'}^{T}(r(\mathbf{x}^*) - r(\mathbf{x}_t))^2}$$

$$\leq \sqrt{\frac{(T - t')^2}{T}\beta C_1 \gamma_T}$$

$$\leq \sqrt{(T - t')\beta C_1 \gamma_T}$$

$\square$

## B.6    Corollary 3

Following the path of proof for Theorem 1, with Lemma 2, we can show that the algorithm can identify infeasibility when all points in the search space violate at least one of the constraints at least $\epsilon'_\mathcal{C}$. Concretely, $\forall \mathbf{x} \in \mathbf{X}$, if it holds that $\exists m \in \mathbf{M}, \mathcal{C}_m(x) < -\epsilon'_\mathcal{C}$, with high probability the identified $\tilde{D}_{\hat{\mathbf{X}}_T} = \emptyset$.

**Corollary 3.** *When the assumptions except for Assumption 2 hold, $\forall \mathbf{x} \in \mathbf{X}$, if $\exists m \in \mathbf{M}, \mathcal{C}_m(x) < -\epsilon'_\mathcal{C}$, then with a constant $\beta = 2\log(\frac{2(M+1)|\tilde{D}|T}{\delta})$ and the acquisition function from Algorithm 2, after at most $T \geq \frac{\beta\widehat{\gamma_T}C_1}{\epsilon'^2_\mathcal{C}}$ iterations, we have $\mathbb{P}\left[\tilde{D}_{\hat{\mathbf{X}}_T} = \emptyset\right] \geq 1 - \delta$. Here, $C_1 = 8/\log(1 + \sigma^{-2})$.*

*Proof.* We assume $\tilde{D}_{\hat{X}_T} \neq \emptyset$ and prove by contradiction. Given equation 6 and 7 and Lemma 2, we have $\forall g \in \{\mathcal{C}_m\}_{m\in\mathbf{M}}$,

$$\max_{\mathbf{x}\in\tilde{D}_{\hat{X}_T}\cap U_{g,T}} \mathrm{UCB}_{g,T}(\mathbf{x}) - \mathrm{LCB}_{g,T}(\mathbf{x}) \leq \epsilon'_\mathcal{C}$$

According to the definition of $U_{g,T}$, $\forall \mathbf{x} \in \tilde{D}_{\hat{X}_T} \cap U_{g,T}, \forall g \in \{\mathcal{C}_m\}_{m\in\mathbf{M}}$, with probability at least $1 - 1/2\delta$, we have

$$\mathcal{C}_m(\mathbf{x}) \leq \mathrm{UCB}_{\mathcal{C}_m,T}(\mathbf{x}) \leq \epsilon'_\mathcal{C} + \mathrm{LCB}_{g,T}(\mathbf{x}) \leq \epsilon'_\mathcal{C} + \mathcal{C}_m(\mathbf{x})$$

Then we have $\forall \mathbf{x} \in \tilde{D}_{\hat{X}_T} \cap U_{g,T}, \exists m \in \mathbf{M}$

$$\mathbb{P}\left[\mathcal{C}_m(\mathbf{x}) \leq \epsilon'_\mathcal{C} + \mathcal{C}_m(\mathbf{x}) < 0\right] \geq 1 - 1/2\delta$$

This contradiction means $\forall g \in \{\mathcal{C}_m\}_{m\in\mathbf{M}}, \tilde{D}_{\hat{X}_T} \cap U_{g,T} = \emptyset$ with probability as least $1 - 1/2\delta$.

According to the definition of $S_{g,T}$, $\forall \mathbf{x} \in \tilde{D}_{\hat{X}_T} \cap S_{g,T}, \forall g \in \{\mathcal{C}_m\}_{m\in\mathbf{M}}$

$$\mathrm{LCB}_{g,T}(\mathbf{x}) \geq \epsilon'_\mathcal{C}$$

Then we have $\forall \mathbf{x} \in \tilde{D}_{\hat{X}_T} \cap S_{g,T}, \exists g \in \{\mathcal{C}_m\}_{m \in \mathbf{M}}$

$$\mathbb{P}\left[-\epsilon'_\mathcal{C} \geq \mathcal{C}_m(\mathbf{x}) \geq \mathrm{LCB}_{g,T}(\mathbf{x}) \geq \epsilon'_\mathcal{C}\right] \geq 1 - 1/2\delta$$

This contradiction means $\forall g \in \{\mathcal{C}_m\}_{m \in \mathbf{M}}$, $\tilde{D}_{\hat{\mathbf{X}}_t} \cap S_{g,T} = \emptyset$ with probability as least $1 - 1/2\delta$.

Combining the above contradictions, we have at least when $t = T$,

$$\mathbb{P}\left[\tilde{D}_{\hat{\mathbf{X}}_T} = \emptyset\right] \geq 1 - \delta$$

$\square$

# C   Continuous Search Space

## C.1   Theoretical Results

In the following, we introduce the additional assumption that bounds the unknown functions' complexity when they are members of an RKHS space and enables the performance analysis when applying COBAR on continuous search space $\mathbf{X}$ instead of $\tilde{D}$.

**Assumption 4.** *The objective and constraints all lie in the RKHS $\mathcal{H}_k$ corresponding to the kernel $k(\mathbf{x}, \mathbf{x}')$, and the corresponding norm is bounded by $\mathcal{B}$. Formally, $f : \mathbf{X} \to \mathbb{R}$ is a member of the RKHS of real-valued functions on $\mathbf{X}$ with kernel $k$, with RKHS norm $\|f\|_k \leq \mathcal{B}$. Similarly, $\mathcal{C}_m : \mathbf{X} \to \mathbb{R}$ is a member of the RKHS of real-valued functions on $\mathbf{X}$ with kernel $k$, with RKHS norm $\|\mathcal{C}_m\|_k \leq \mathcal{B}$, for all $m \in \mathbf{M}$.*

Then, we could derive similar results mapping from Lemma 1.

*Lemma* 1.  Under the assumptions above, the regions of interest $\hat{\mathbf{X}}_t$, as defined in equation 2, contain the global optimum with high probability. Formally, for all $\delta \in (0, 1)$, $T \geq t \geq 1$, and the search space $\mathbf{X}$ that contains the optimum $\mathbf{x}^* = \arg\max_{\mathbf{x} \in \mathbf{X}} f(\mathbf{x})$ where $\mathcal{C}_m(\mathbf{x}^*) > \epsilon_\mathcal{C}$ for all $m \in \mathbf{M}$ and $\beta_t^{1/2} = B + \sigma\sqrt{2(\widehat{\gamma_T} + 1 + ln(2(M+1)/\delta))}$, we have $\mathbb{P}\left[\mathbf{x}^* \in \hat{\mathbf{X}}_t\right] \geq 1 - \delta$.

*Proof.*  Similar to theorem 2 of Chowdhury and Gopalan [2017], with probability at least $1 - 1/2\delta$, $\forall \mathbf{x} \in \tilde{D}, \forall T \geq t \geq 1, \forall g \in \{f\} \cup \{\mathcal{C}_m\}_{m \in \mathbf{M}}$,

$$|g(\mathbf{x}) - \mu_{g,t-1}(\mathbf{x})| \leq \beta_t^{1/2}\sigma_{g,t-1}(\mathbf{x})$$

Note that we also take the union bound on $g \in \{f\} \cup \{\mathcal{C}_m\}_{m \in \mathbf{M}}$.

First, by definition $S_{\mathcal{C},t} \triangleq \bigcap_m^{\mathbf{M}} S_{\mathcal{C}_m,t}$, we have $\forall t \leq T, \mathbf{x} \in S_{\mathcal{C},t}, \forall m \in \mathbf{M}$

$$\mathbb{P}\left[\mathcal{C}_m(\mathbf{x}) \geq \mathrm{LCB}_{\mathcal{C}_m,t}(\mathbf{x}) > 0\right] \geq 1 - 1/2\delta$$

meaning with probability at $1 - \delta$, $\mathbf{x}$ lies in the feasible region. At the same time, we have, $\forall t \leq T$, $\forall m \in \mathbf{M}$, given $\mathcal{C}_m(\mathbf{x}) > 0$

$$\mathbb{P}\left[\mathrm{UCB}_{f,t}(\mathbf{x}^*) \geq f(\mathbf{x}^*) \geq f(\mathbf{x}) \geq \mathrm{LCB}_{f,t}(\mathbf{x})\right] \geq 1 - 1/2\delta$$

Given the mutual independency between the objective $f$ and the constraints $\mathcal{C}_m$, and by the definition of the threshold $\mathrm{LCB}_{f,t,\max}$, we have $\forall t \leq T$, when $\exists \mathbf{x} \in S_{\mathcal{C},t}$,

$$\mathbb{P}\left[\mathrm{UCB}_{f,t}(\mathbf{x}^*) > \mathrm{LCB}_{f,t,\max}\right] \geq 1 - \delta$$

Note when $S_{\mathcal{C},t} = \emptyset$, $\mathrm{LCB}_{f,t,\max} = -\infty$, we have $\mathbb{P}\left[\mathrm{UCB}_{f,t}(\mathbf{x}^*) > \mathrm{LCB}_{f,t,\max}\right] = 1$.

In summary, we've shown that with probability at least $1 - \delta$, $\mathbf{x}^* \in \hat{\mathbf{X}}_{f,t}$.

Next, by the definition of $\mathbf{x}^* = \arg\max_{\mathbf{x} \in \mathbf{X}} f(\mathbf{x})$ $s.t.$ $\mathcal{C}_m(\mathbf{x}^*) > \epsilon_\mathcal{C}$ we have $\forall t \leq T, \forall m \in \mathbf{M}$

$$\mathbb{P}\left[\mathrm{UCB}_{\mathcal{C}_m,t}(\mathbf{x}^*) \geq \mathcal{C}_m(\mathbf{x}^*) > 0\right] \geq 1 - 1/2\delta$$

meaning with probability at least $1 - 1/2\delta$, $\mathbf{x}^* \in \hat{\mathbf{X}}_{\mathcal{C}_m,t}$. And in general, we have $\forall t \leq T, \forall m \in \mathbf{M}$

$$\mathbb{P}\left[\mathbf{x}^* \in \hat{\mathbf{X}}_t\right] \geq 1 - \delta$$

$\square$

*Remark* 2. The proof of Lemma 1 substitutes the $\beta$ in the proof of Lemma 1 and alleviates the need for a discretization $\tilde{D}$ with the additional assumption on the complexity of the unknown functions in Assumption 4. Note the $\beta_t^{1/2} = B + \sigma\sqrt{2(\widehat{\gamma_T} + 1 + ln(2(M+1)/\delta))}$, we have $\mathbb{P}\left[\mathbf{x}^* \in \hat{\mathbf{X}}_t\right] \geq 1 - \delta$ is larger than the original value in the theorem 2 of Chowdhury and Gopalan [2017] to make sure $\beta_t$ is the same for all $\forall T \geq t \geq 1$ and to guarantee a union bound on $g \in \{f\} \cup \{\mathcal{C}_m\}_{m \in \mathbf{M}}$. In the following, since $\beta_t$ is constant, we substitute it with $\beta$.

Then, we could trivially map the Theorem 1 when maximizing the acquisition functions on $\hat{\mathbf{X}}_t$ instead of $\tilde{D}_{\hat{\mathbf{X}}_t}$ as in line 9 of Algorithm 2 and on $\hat{\mathbf{X}}_t \cap U_{\mathcal{C}_m,t}$ instead of $\tilde{D}_{\hat{\mathbf{X}}_t} \cap U_{\mathcal{C}_m,t}$ as in line 8 of Algorithm 2. The proof would be identical to section B except for the different $\beta$ and search space.

---

**Algorithm 1 CO**nstrained **B**O through **A**daptive **R**egion of Interest Acquisition on Continuous Space(COBAR-CS)

---

1: **Input:**Search space $\mathbf{X}$, initial observation $\mathbf{S}_0$, horizon $T$, confidence factor $\delta$, confidence coefficient $\beta$;
2: **for** $t = 1\ to\ T$ **do**
3:     Update the posteriors of $\mathcal{GP}_{f,t}$ and $\mathcal{GP}_{\mathcal{C}_m,t}$
4:     Identify ROIs $\hat{\mathbf{X}}_t$, and undecided sets $U_{\mathcal{C}_m,t}$
5:     **for** $m \in \mathbf{M}$ **do**
6:       **if** $U_{\mathcal{C}_m,t} \neq \emptyset$ **then**
7:         Candidate for learning of each constraint:
        $\mathbf{x}_{\mathcal{C}_m,t} \leftarrow \arg\max_{\mathbf{x} \in \hat{\mathbf{X}}_t \cap U_{\mathcal{C}_m,t}} \alpha_{\mathcal{C}_m,t}(\mathbf{x})$ (4)
8:         $\mathcal{G} \leftarrow \mathcal{G} \cup \mathcal{C}_{m,t}$
9:     Candidate for optimizing the objective:
    $\mathbf{x}_{f,t} \leftarrow \arg\max_{\mathbf{x} \in \hat{\mathbf{X}}_t} \alpha_{f,t}(\mathbf{x})$ as in equation 3
10:     $\mathcal{G} \leftarrow \mathcal{G} \cup f$
11:     Maximize the acquisition from different aspects:
    $g_t \leftarrow \arg\max_{g \in \mathcal{G}} \alpha_{g,t}(\mathbf{x}_{g,t})$
12:     Pick the candidate to evaluate: $\mathbf{x}_t \leftarrow \mathbf{x}_{g_t,t}$
13:     Update the observation set
    $\mathbf{S}_t \leftarrow \mathbf{S}_{t-1} \cup \{(\mathbf{x}_t, y_{f,t}, \{y_{\mathcal{C}_m,t}\}_{m \in \mathbf{M}})\}$

---

**Theorem 3.** *Under the aforementioned assumptions, with a constant $\beta_t^{1/2} \triangleq \beta^{1/2} = B + \sigma\sqrt{2(\widehat{\gamma_T} + 1 + ln(2(M+1)/\delta))}$ and the acquisition function from $Algorithm$ 1, there exists an $\epsilon_f \leq \epsilon_{\mathcal{C}}$, such that after at most $T \geq \frac{\beta\widehat{\gamma_T}C_1}{\epsilon_f^2}$ iterations, we have $\mathbb{P}\left[|CI_{f^*,T}| \leq \epsilon_f, f^* \in CI_{f^*,T}\right] \geq 1 - \delta$ Here, $C_1 = 8/\log(1 + \sigma^{-2})$.*

### C.2 Efficient Discretization

Aiming at a continuous search space demands additional consideration when implementing a practical ROI identification on the continuous search space or requires a better coverage by the discretization of the dense search space for COBAR in practice. This problem is more outstanding in high-dimensional tasks. Here, we briefly discuss potential remedies if we still resort to an efficient discretization. The random linear projection has been used for discretizing the search space to mitigate the dependency on the dimensionality while, with high probability, preserving the original geometry [Dasgupta, 1999, Nayebi et al., 2019]. To efficiently discretize the dense search space for COBAR (in high-dimensional applications), one option is to apply the random projection and its reverse studied by Nayebi et al. [2019], which shows strong empirical performance when combined with other BO algorithms and offers the following theoretical guarantee.

**Definition 1.** *($\varepsilon$-subspace embedding [Nayebi et al., 2019]) Given a matrix $V \in \mathbb{R}^{D \times d}$ with orthonormal columns, an integer $d \leq D$ and an approximation parameter $\varepsilon \in (0,1)$, an $\varepsilon$-subspace embedding for $V$ is a map $H : \mathbb{R}^d \to \mathbb{R}^D$ such that $\forall \mathbf{x} \in \mathbb{R}^d$:*

$$(1 - \varepsilon)\|V\mathbf{x}\|_2^2 \leq \|HV\mathbf{x}\|_2^2 \leq (1 + \varepsilon)\|V\mathbf{x}\|_2^2$$

| Problem | COBAR | CMES-IBO | SCBO | cEI |
|---|---|---|---|---|
| Ackley-40D-2C | 75.87 | 281.47 | 34.33 | 185.07 |

Table 1: Average wall time (sec) of different CBO Methods collected from 15 independent trials.

**Theorem 4.** *(Theorem 2 of Nayebi et al. [2019]) Consider a Gaussian process that acts directly in the unknown active subspace of dimension $d_e$ with mean and variance functions $\mu(\cdot), \sigma^2(\cdot)$. Let $\hat{\mu}(\cdot), \hat{\sigma}^2(\cdot)$ be their approximations using an $\varepsilon$-subspace embedding for the active subspace. Then we have for every $\mathbf{x} \in X$*

*1. $|\mu(\mathbf{x}) - \hat{\mu}(\mathbf{x})| \leq 5\varepsilon \|\mathbf{x}\| \|X - \hat{f}\|$*

*2. $\sigma^2(\mathbf{x}) - \hat{\sigma}^2(\mathbf{x}) \leq 12\varepsilon \|\mathbf{x}\|^2$*

For a comprehensive survey on the treatments of high-dimensional search space for BO, we refer to the recent survey by Binois and Wycoff [2022]. Besides the random projection [Nayebi et al., 2019, Wang et al., 2016, Letham et al., 2020], variable selection Hellsten et al. [2023], tree-structure partition [Eriksson and Jankowiak, 2021], and Markov Chain Monte Carlo sampling [Yi et al., 2024] on the search space could all be applied as plugins to improve the discretization efficiency for COBAR.

### C.3 Case Study

We illustrate the effectiveness of integrating Hashing-enhanced Subspace BO (HeSBO) [Nayebi et al., 2019] into CBO algorithms. We construct the following 40-dimensional CBO task that makes any grid discretization containing a feasible candidate on the original embedding intractable.

**Ackley-40D-2C** $f(\mathbf{x}) = 20 \exp\left(-0.2\sqrt{1/d \sum_i^d x_i^2}\right) + \exp\left(1/d \sum_i^d \cos(2\pi x_i)\right) + 20 + \exp(1)$, $d = 5$ where $\mathbf{x} \in [-5, 10]^{40}$. We construct two constraints to enforce a feasible area taking up less than $0.6\%$ of the search space. The first constraint $\mathcal{C}_1 = 1 - (\sum_i^5 x_i)$. The second constraint $\mathcal{C}_2 = 6 - (\sum_i^5 x_i^2)$.

We find that COBAR fails in the original search space $[-5, 10]^{40}$ due to the intractability of any discretization containing sufficient feasible candidates. We integrate HeSBO into all the tested CBO algorithms to allow the algorithms to process on a 5-dimensional embedding space $[-1, 1]^5$. COBAR relies on a random sampling containing 200000 candidates. The simple regret curves are shown in figure 4. Though the point-wise comparison of COBAR is not tractable in the original 40-dimensional search space, integrating HeSBO allows COBAR to optimize the high-dimensional CBO toy problem efficiently. Table 1 shows that COBAR could efficiently optimize the embedding space, benefited from the reduced dimensionality and the ROI identification that further reduces computation need dynamically.

## D  Dealing with Boundary Optimum

Here, we discuss the treatment and theoretical behavior when dealing with the boundary optimum. First, we extend the results in Theorem 1, when not assuming the Assumption 2 hold. We uniformly shift the constraints by a small amount $\epsilon_{\mathcal{C}}$ to satisfy Assumption 2 with the modified constraints. Formally, $\forall m \in \mathbf{M}, \mathcal{C}'_m(\mathbf{x}) = \mathcal{C}_m(\mathbf{x}) + \epsilon_{\mathcal{C}}$. Then, running COBAR with these adjusted constraints, $\mathcal{C}'_m$, instead of the original $\mathcal{C}_m$, we have the following guarantee, which is a direct extension of Theorem 1. We denote the $\tilde{f}^* = f(\tilde{\mathbf{x}}^*)$, Here $\tilde{\mathbf{x}}^* = \arg\max_{\mathbf{x} \in \mathbf{X}, \forall m \in \mathbf{M}, \mathcal{C}'_m(\mathbf{x}) > 0} f(\mathbf{x})$.

**Corollary 4.** *Under the aforementioned assumptions and modifications, with a constant $\beta = 2\log(\frac{2(M+1)|\tilde{D}|T}{\delta})$ and the acquisition function from $Algorithm$ 2, there exists an $\epsilon_f \leq \epsilon_{\mathcal{C}}$, such that after at most $T \geq \frac{\beta \hat{\gamma}_T C_1}{\epsilon_f^2}$ iterations, we have $\mathbb{P}\left[|CI_{\tilde{f}^*, T}| \leq \epsilon_f, \tilde{f}^* \in CI_{\tilde{f}^*, T}\right] \geq 1 - \delta$. Here, $C_1 = 8/\log(1 + \sigma^{-2})$.*

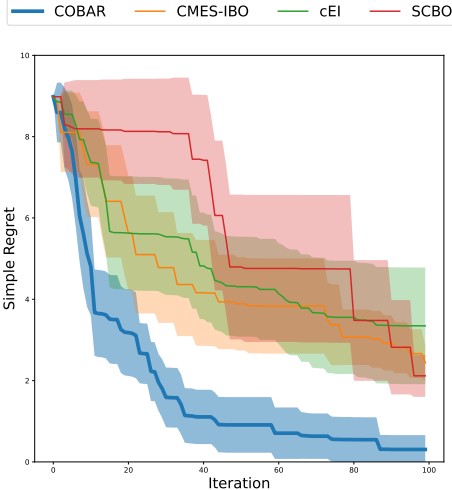

Figure 4: The figure illustrates the simple regret for Ackley-40D-2C. All the tested algorithms rely on the low-dimensional embedding in HeSBO. The results are collected from 15 independent trials. The shaded area denotes the standard error.

This Corollary 4 allows us to depict the width of the global optimum defined in the enlarged feasible region similarly. Since the optimum is defined in the enlarged area, it could be an upper bound of the global optimum defined in the original feasible region, including the feasible region boundaries. That is

$$\tilde{f}^* \geq \underset{\mathbf{x}\in\mathbf{X},\forall m\in\mathbf{M},\mathcal{C}_m(\mathbf{x})\geq 0}{\arg\max} f(\mathbf{x})$$

This allows us to extend further the Corollary 2 that depicts the partial cumulative regret after sufficient iterations and the upper bound of the violations.

**Corollary 5.** *Under the aforementioned assumptions and modifications, when $\forall m \in \mathbf{M}$, $\mathcal{C}'_m(\mathbf{x}) > 0$, $\mathbf{x} \neq \mathbf{x}^*$, $\exists \epsilon_\mathcal{C} > \epsilon'_\mathcal{C} \geq \epsilon_f > 0$, $f^* - f(\mathbf{x}) \leq 2\epsilon_f$, it holds that $\forall m \in \mathbf{M}$, $\mathcal{C}'_m(\mathbf{x}) \geq \epsilon'_\mathcal{C}$. We use $\beta = 2\log(\frac{2(M+1)|\tilde{D}|T}{\delta})$ and the acquisition function from Algorithm 2. After at most $t' \geq \frac{\beta\widehat{\gamma_{t'}}C_1}{\epsilon'^2_f}$ iterations, we have, $\mathbb{P}\left[\sum_{t=t'}^{T} r(\mathbf{x}^*) - r(\mathbf{x}_t) \leq \sqrt{(T-t')\beta\gamma_T C_1}, \forall \mathbf{x} \in \tilde{D}, \mathcal{C}_m(\mathbf{x}_t) \geq -\epsilon_\mathcal{C}\right] > 1 - \delta$. Here, $C_1 = 8/\log(1 + \sigma^{-2})$ and $t' \leq T$.*

*Proof.* First, we have $\forall \mathbf{x} \in \tilde{D}_{\hat{\mathbf{X}}_t}$, $2\beta^{1/2}\sigma_{f,t-1}(\mathbf{x}) \leq \alpha_t \leq \epsilon_f$.

Then $\forall \mathbf{x} \neq \tilde{\mathbf{x}}^*$ and $\mathbf{x} \in \tilde{D}_{\hat{\mathbf{X}}_t}$, we have

$$\mathbb{P}\left[\tilde{f}^* - f(\mathbf{x}) \leq |CI_{\tilde{f}^*,t}| + \text{UCB}_{f,t}(\mathbf{x}) - \text{LCB}_{f,t}(\mathbf{x}) \leq 2\alpha_t \leq 2\epsilon_f\right] \geq 1 - \delta$$

Then by assumption, $\forall \mathbf{x} \in \tilde{D}_{\hat{\mathbf{X}}_t}$, $\forall m \in \mathbf{M}$, we have probability at least $1 - \delta$, $\mathcal{C}'_m(\mathbf{x}) \geq \epsilon'_\mathcal{C}$. Hence we have both $\mathcal{C}_m(\mathbf{x}) \geq \epsilon_\mathcal{C} - \epsilon'_\mathcal{C}$ and $\mathbf{x} \notin U_{\mathcal{C}'_m,t}$. According to the algorithm, it regresses to GP-UCB by Srinivas et al. [2009] between $t'$ and $T$.

$$\begin{aligned}\sum_{t=t'}^{T}(r(\mathbf{x}^*) - r(\mathbf{x}_t)) &\leq \sum_{t=t'}^{T}(r(\tilde{\mathbf{x}}^*) - r(\mathbf{x}_t))^2 \\ &\leq \beta C_1(\gamma_T - \gamma_{t'}) \\ &\leq \beta C_1 \gamma_T(1 - t'/T)\end{aligned}$$

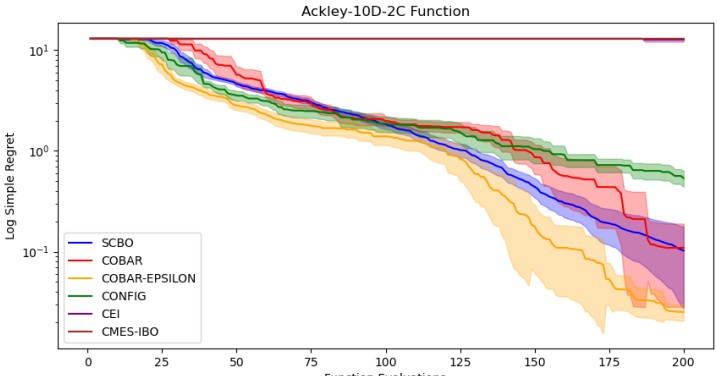

Figure 5: The figure illustrates the simple regret for Ackley-10D-2C. The results are collected from 15 independent trials. The shaded area denotes the 98% confidence interval. We reproduce the reported performance of SCBO using the corresponding Botorch tutorial. Then, we fix the kernel choices and other hyperparameters to make a fair comparison. For COBAR-EPSILON, we set $\epsilon_{\mathcal{C}} = 1.2$.

By Cauchy-Schwarz, we have

$$\sum_{t=t'}^{T}(r(\mathbf{x}^*) - r(\mathbf{x}_t)) \leq \sqrt{(T-t')\sum_{t=t'}^{T}(r(\mathbf{x}^*) - r(\mathbf{x}_t))^2}$$

$$\leq \sqrt{\frac{(T-t')^2}{T}\beta C_1 \gamma_T}$$

$$\leq \sqrt{(T-t')\beta C_1 \gamma_T}$$

$\square$

Note that by enlarging the feasible region with $\epsilon_{\mathcal{C}}$, we don't risk losing the feasible region to enable COBAR to identify both interior and boundary optimum. We don't change the definition of $f^*$. Instead, we only leverage the modified running constraints and verify the feasibility with the original constraints. For the algorithms aiming at violation tolerant objectives like CONFIG [Xu et al., 2023], there is no similar guarantee with straightforward modification, e.g., adding small $\epsilon_{\mathcal{C}}$ to the threshold while not risking losing the feasible region. We denote the modified COBAR as COBAR-EPSILON. We include the corresponding comparison on the noise-free Ackley-10D-2C as exactly defined in Eriksson and Poloczek [2021], where the feasible region is less than $2.2 * 10^{-3}\%$ of the whole search space, and the optimum lies on the boundary of feasible region by construction.

As is shown in figure 5, COBAR is initially outperformed by SCBO while converging to the near-optimal area after the sufficient budget as SCBO. COBAR-EPSILON archives the best convergence throughout the optimization with the proposed minor tweak. In contrast, CONFIG fails to converge to the global optimum, possibly due to its tolerance of the constraints violation, and by definition, no reward is incurred for a point outside the feasible region.

## E    Reward Function

### E.1    Reward choice 1: product of reward and feasibility

The definition of reward plays an important role in online machine learning performance analysis. In the CBO setting, one possible definition of constrained reward derived from the constraint nature is $r(\mathbf{x}) = f(\mathbf{x})\prod_m \mathbb{I}_{\mathcal{C}_m(\mathbf{x}) \geq h_m}$ when assuming the $f(\mathbf{x}) > 0$. Considering both the aleatoric and epistemic uncertainty on the constraints, we could transform the problem into finding the maximizer

$$\arg\max_{\mathbf{x}\in\mathbf{X}} r(\mathbf{x}) = \arg\max_{\mathbf{x}\in\mathbf{X}} f(\mathbf{x})\prod_m \mathbb{P}\left[Y_{\mathcal{C}_m}(\mathbf{x}) \geq h_m\right]$$

Here $Y_{\mathcal{C}_m}(\mathbf{x})$ denotes the observation of the constraint $\mathcal{C}_m$ at $\mathbf{x}$.

The problem with this product reward, on the one hand, is that it is likely to incur a Pareto front if we regard the problem as a multi-objective optimization where the objectives are composed of $f(\mathbf{x})$ and $\mathbb{P}\left[Y_{\mathcal{C}_m}(\mathbf{x}) \geq h_m\right]$. The multi-objective nature and resulting Pareto front indicate that the optimization could be more challenging to converge than the single-objective unconstrained BO problem, though the unique global optimum is not always expected there either. More critically, when the feasibility of reaching a certain threshold, we prefer to focus on optimizing the objective value rather than the product for the following reasons.

Firstly, the marginal gain on improving feasibility by increasing the value of the constraint function drops after the feasibility reaches 0.5 assuming it follows a Gaussian. Especially in the tail region, improving the feasibility and then the product of feasibility and objective value by optimizing the constraint function is prohibitively difficult.

Secondly, in most real-world scenarios except for certain applications that focus on feasibility (where the feasibility should be treated as another objective and make it in nature a multi-objective optimization), the actual marginal gain, in general, increases the feasibility decay faster than the increase of objective value. (e.g., when choosing between doubling the feasibility from 0.25 to 0.5 or doubling the objective value drop from 25 to 50, we probably favor the former as 0.25, meaning it is unlikely to happen. However, when choosing between increasing feasibility from .8 to .9 or increasing the objective drop from 80 to 90, there would be no such clear preference.) Then, the user would possibly favor the gain on the objective function after the feasibility reaches a certain level. Therefore, we propose the following reward for constrained optimization tasks according to this insight.

### E.2 Reward choice 2: objective function after the feasibility reaching certain threshold

Instead of defining the reward as the product of the objective value and feasibility, we have to look into the probabilistic constraints and distinguish the epistemic uncertainty and aleatoric uncertainty. First, when assuming the observation on the constraints are noise-free, namely $Y_{\mathcal{C}_m}(\mathbf{x}) = \mathcal{C}_m(\mathbf{x})$, we could simply use the indicator function $\mu_m$ for each constraint to turn the feasibility function into an indicator function. This definition accommodates the scenarios where the infeasible region does not incur credible reward as discussed by Sacher et al. [2018], Bachoc et al. [2020] due to simulation failures

$$r(\mathbf{x}) = \begin{cases} f(\mathbf{x}) & if \quad \mathbb{I}(C_m(\mathbf{x}) \geq h_m) \quad \forall m \in \mathbf{M} \\ -inf & o.w \end{cases} \tag{11}$$

Next, if the observation on the constraints is perturbed with a known Gaussian noise, namely $Y_{\mathcal{C}_m}(\mathbf{x}) \sim \mathcal{N}(\mathcal{C}_m(\mathbf{x}), \sigma)$, we could deal with the aleatoric uncertainty with a user-specific confidence level for each constraint $\chi_m \in (0,1), \forall m \in \mathbf{M}$. Then we could turn $\mathbb{I}(Y_{C_m}(\mathbf{x}) \geq h_m)$ into probabilistic constraints following the definiation proposed by Gelbart et al. [2014] and

$$\mathbb{P}\left[Y_{C_m}(\mathbf{x}) \geq h_m\right] \geq \chi_m$$

to explicitly deal with the aleatoric uncertainty. With the percentage point function (PPF), we could transform the probabilistic constraints into a deterministic constraint $\mathbb{I}(C_m(\mathbf{x}) \geq \hat{h}_m)$ with $\hat{h}_m = PPF(h_m, \sigma, \mu_m)$, meaning $\hat{h}$ is the $\chi_m$ percent point of a Gaussian distribution with $h_m$ and $\sigma$ as its mean and standard deviation. Hence, we could unify the form of rewards of noise-free and noisy observation on the constraints with the user-specified confidence levels. For simplicity and without loss of generalization, we stick to the definition in equation 1 and let all $\hat{h}_m = 0$.

## F   Additional Experiment Details

In the benchmarks, we applied the deep Gaussian process [Wilson et al., 2016]. Here, we offer a more detailed discussion of the construction of the six CBO tasks studied in section 4.

### F.1 Synthetic Tasks

We study two synthetic CBO tasks constructed from conventional BO benchmark tasks. Here, we rely on the implementation contained in BoTorch's [Balandat et al., 2020] test function module.

**Rastrigin-1D-1C**  The Rastrigin function is a non-convex function used as a performance test problem for optimization algorithms. It was first proposed by Rastrigin [1974] and used as a popular benchmark dataset [Pohlheim, 2006]. It is constructed to be highly multimodal, with local optima being regularly distributed to trap optimization algorithms. Concretely, we negate the 1D Rastrigin function and try to find its maximum: $f(\mathbf{x}) = -10d - \sum_{i=1}^{d} (x_i^2 - 10\cos(2\pi x_i))$, $d = 1$. The range of $\mathbf{x}$ is $[-5, 5]$, and we construct the constraint to be $c(\mathbf{x}) = |\mathbf{x} + 0.7|^{1/2}$. When setting the threshold as $\sqrt{2}$, we essentially exclude the global optimum from the feasible area. The constraint enforces the optimization algorithm to explore feasibility rather than allowing algorithms to improve the reward by merely optimizing the objective. Then, the feasible region takes up approximately 60% of the search space. This one-dimensional task is designed to illustrate the necessity of adaptively trade-off learning of constraints and optimization of the objective.

We also vary the threshold to control the portion of the feasible region to study the robustness of COBAR. Figure 6 shows the distribution of the objective function and feasible regions on the samples.

**Ackley-5D-2C**  The Ackley function is also a popular benchmark for optimization algorithms. Compared with the Rastrigin function, it is similarly highly multimodal, while the region near the center is growingly steep. Same as what is done for Rastrigin, we negate the 5D Ackley function and try to find its maximum: $f(\mathbf{x}) = 20\exp\left(-0.2\sqrt{1/d \sum_i^d x_i^2}\right) + \exp\left(1/d \sum_i^d \cos(2\pi x_i)\right) + 20 + \exp(1)$, $d = 5$. The search space is restricted to $[-5, 3]^5$. We construct two constraints to enforce a feasible area approximately taking up 14% of the search space. The first constraint $(\|x - \mathbf{1}\|_2 - 5.5)^2 - 1$ constructs two feasible regions. One of them lies in the center, and the other is close to the boundary of the search space. The second constraint $-\|x\|_\infty^2 + 9$ allows one hypercube feasible region in the center.

### F.2 Real-world Tasks

We study four real-world CBO tasks. The first three are extracted from Tanabe and Ishibuchi [2020], which offers a broad selection of real-world multi-objective multi-constraints optimization tasks. The fourth one is a 32-dimensional optimization task extracted from the UCI Machine Learning repository [mis, 2019].

**Vessel-4D-3C**  The pressure vessel design problem aims to optimize the total cost of a cylindrical pressure vessel. The four variables represent the thicknesses of the shell, the head of a pressure vessel, the inner radius, and the length of the cylindrical section. The problem is originally studied in Kannan and Kramer [1994], and we follow the formulation in RE2-4-3 in Tanabe and Ishibuchi [2020]. The feasible regions take up approximately 78% of the whole search space.

**Spring-3D-6C**  The coil compression spring design problem aims to optimize the volume of spring steel wire, which is used to manufacture the spring [Lampinen and Zelinka, 1999] under static loading. The three input variables denote the number of spring coils, the outside diameter of the spring, and the spring wire diameter, respectively. The constraints incorporate the mechanical characteristics of the spring in real-world applications. We follow the formulation in RE2-3-5 in Tanabe and Ishibuchi [2020]. The feasible regions take up approximately 0.38% of the whole search space.

**Car-7D-8C**  The car cab design problem includes seven input variables and eight constraints. The problem is originally studied in Deb and Jain [2013]. We follow the problem formulation in RE9-7-1 in Tanabe and Ishibuchi [2020] and focus on the objective of minimizing the weight of the car while meeting the European enhanced Vehicle-Safety Committee (EEVC) safety performance constraints. The seven variables indicate the thickness of different parts of the car. The feasible region takes up approximately 13% of the whole search space.

**Converter-32D-3C** This UCI dataset we use consists of positions and absorbed power outputs of wave energy converters (WECs) from the southern coast of Sydney. The applied converter model is a fully submerged three-tether converter called CETO. 16 WECs 2D-coordinates are placed and optimized in a size-constrained environment [mis, 2019]. The input is, therefore, 32 dimensional. We place three constraints on the tasks, including the absorbed power of the first two converters being above a certain threshold of 96000 and the general position being not too distant with the two-norm below 2000. The feasible region takes up approximately 27% of the whole search space.

# G    Additional Experiments

Here, we provide additional experiment results on COBAR.

## G.1    Different Portion of Feasible Region for Rastrigin-1D

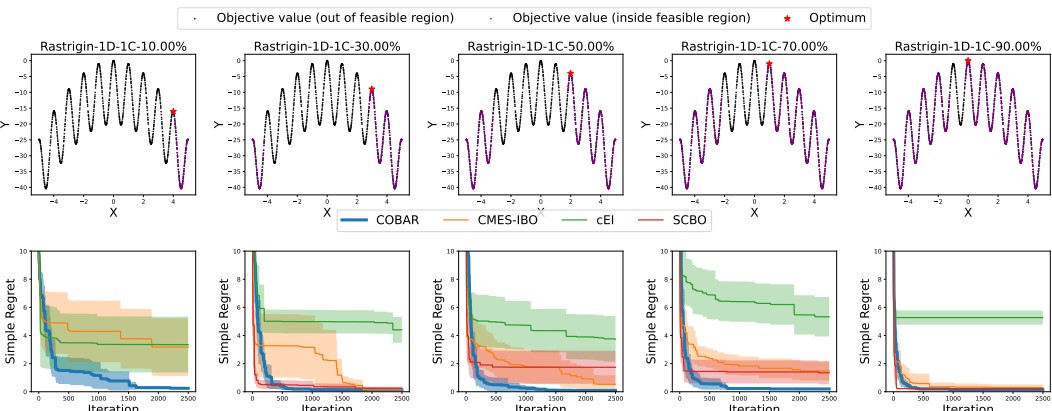

Figure 6: We use black dots and purple dots to show the infeasible region and feasible region in the first row correspondingly. Each column corresponds to a certain threshold choice for the single constraint $c(\mathbf{x}) = |\mathbf{x} + 0.7|^{1/2}$ in the Rastrigin-1D-1C task. The search space contains a certain portion of the feasible region, denoted on each figure and title. The first row shows the distribution of 1000 samples from the noise-free distribution objective function, and the figures are differentiated with different feasible regions. The second row shows corresponding simple regret curves. We test each method with 15 independent trails and impose observation noises sampled from $\mathcal{N}(0, 0.1)$ not shown in the first row. The scaling and length scale of the GPs are learned via maximum likelihood estimation.

## G.2    Robustness to Choices of $\beta$

As is shown in figure 7, the algorithm is robust to moderate values of $\beta$. Except from the Ackley $\beta = 0.1$ where the filtering of ROI is over-aggressive and traps the model on a certain locality when a very small number of candidates remain in ROI. We observe that certain $\beta$ choices could be slightly better but don't impact the convergence and lack statistical significance. We believe the acquisitions in Eq. (4) and Eq. (3), together with the $\hat{\mathbf{X}}$ identification when the models are well-fitted, contribute to this robustness. Different from conventional GP-UCB [Srinivas et al., 2009], the acquisition functions are standardized with the (maximum) lower confidence bound. The search domains are filtered when historical observations suggest poor performance in nearby areas.

## G.3    Wall Time

We show the wall time of COBAR compared with the baselines in table 2. The results demonstrate the efficiency of COBAR due to the ROI filtering reducing the search space, though the ROI identification incurs additional cost for membership check.

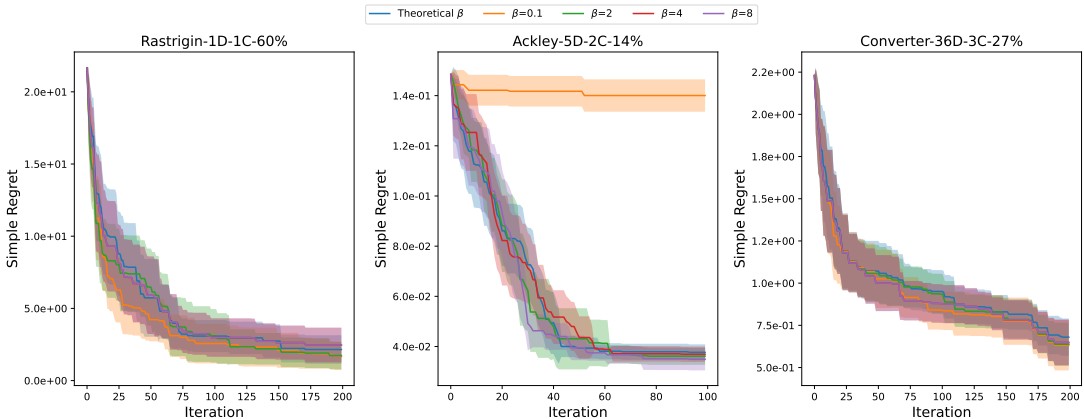

Figure 7: The figure illustrates the simple regret for a different choice of constant $\beta$ for COBAR. Here the theoretical $\beta$ are 6.51 for Rastrigin-1D-1C, 6.47 for Ackley-5D-2C, and 6.51 for Converter-36D-3C. The results are collected from 15 independent trials.

| Problem | COBAR | CMES-IBO | SCBO | cEI |
|---------|-------|----------|------|-----|
| Rastrigin-1D-1C | 144.29 | 545.83 | 32.39 | 231.12 |
| Ackley-5D-2C | 96.19 | 565.10 | 25.43 | 180.39 |
| Converter-36D-3C | 190.05 | 660.27 | 31.73 | 267.36 |

Table 2: Average wall time (sec) of different CBO Methods collected from 15 independent trials.

### G.4 Additional Comparison with CONFIG

Though the objective is defined differently, we add additional baseline CONFIG from Xu et al. [2023]. The results are shown in figure 8 and figure 9. We observe that COBAR outperforms or at least matches CONFIG in all the problems in our setting. Specifically, in the early stage of the Rastrigin-1D-1C task and through the Ackley-5D-2C, where the underlying objective is highly fluctuating, as is shown in figure 8 for Rastrigin-1D-1C, CONFIG fails to enter the feasible region consistently even after exhausting sufficient budget and gets stuck in learning the constraints passively.

At the same time, we observe that on Converter-32D-3C, Vessel-4D-3C, and Spring-3D-6C, CONFIG generally matches the performance of COBAR. We hypothesize that in these applications, the constraint learning part of COBAR is not as beneficial as directly optimizing the underlying function is possibly feasible regions, as the unknown feasibility coincides with the optimality of the underlying objectives. Still, COBAR bears higher consistency in all the benchmarks, highlighting the efficiency and necessity of the adaptive trade-off of active learning and optimization in COBAR when assuming no reward is incurred outside the feasible region. This difference also highlights the necessity of actively learning the complex underlying constraints to guarantee a stable convergence to a feasible optimum.

### G.5 Additional Comparison with SVM-CBO

SVM-CBO [Antonio, 2021] offers a practicality-oriented solution. It uses the SVM to learn the feasibility bound to estimate the decision boundary efficiently. The challenge of analyzing the learning of SVM combined with the coverage-oriented first-phase acquisition function poses a challenge to regret analysis. In addition, SVM-CBO requires a specific split of feasibility identification and optimization within the feasible region and demands different performance metrics for evaluation. This split makes direct comparisons with COBAR, which is somewhat challenging and does not explicitly split the two processes. Nonetheless, we follow the practice in the paper that uses a 10:60:30 split for the random sampling, phase 1 and phase 2 of SVM-CBO, and report the simple regret.

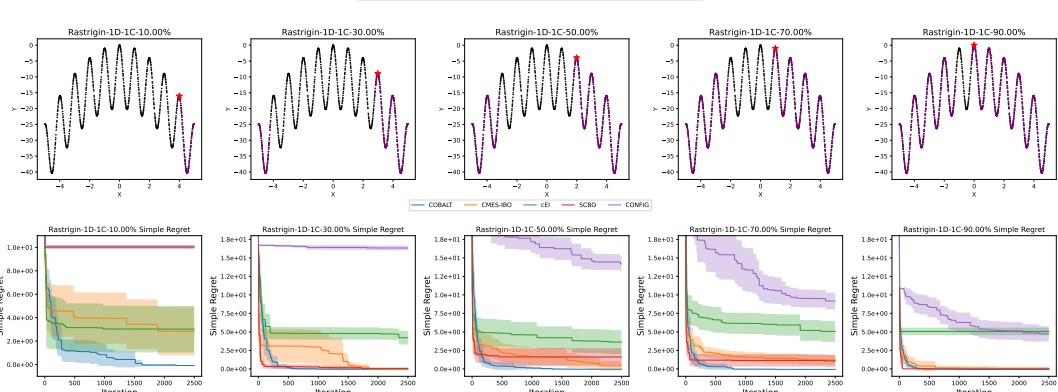

Figure 8: We use black dots and purple dots to show the infeasible region and feasible region in the first row correspondingly. Each column corresponds to a certain threshold choice for the single constraint $c(\mathbf{x}) = |\mathbf{x} + 0.7|^{1/2}$ in the Rastrigin-1D-1C task. The search space contains a certain portion of the feasible region, denoted on each figure and title. The first row shows the distribution of 1000 samples from the noise-free distribution objective function, and the figures are differentiated with different feasible regions. The second row shows corresponding simple regret curves. We test each method with 15 independent trails and impose observation noises sampled from $\mathcal{N}(0, 0.1)$ not shown in the first row. The scaling and length scale of the GPs are learned via maximum likelihood estimation.

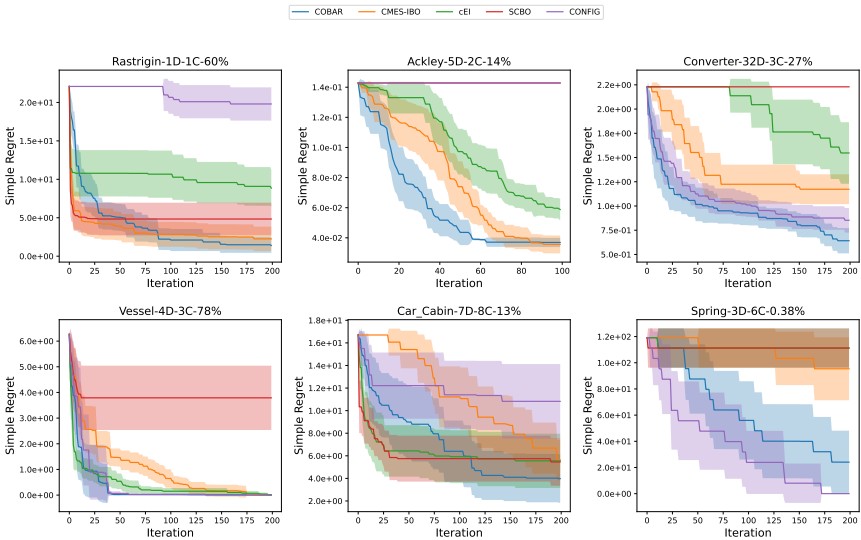

Figure 9: The input dimensionality, the number of constraints, and the approximate portion of the feasible region in the whole search space for each task are denoted on the titles. The curves show the average simple regret after standardization, while the shaded area denotes the 95% confidence interval through the optimization.

Table 3 shows the simple regret of the end of both phases of SVM-CBO. We emphasize the best simple regret achieved. The results demonstrate that COBAR ultimately outperforms or matches the best baseline in the end.

## H  Additional Explanation of COBAR

| Experiment | COBAR-70 | CMES-IBO-70 | cEI-70 | SCBO-70 | SVM-CBO-70 |
|---|---|---|---|---|---|
| Rastrigin-1D-1C-60% | 3.80e+00 (1.85e+00) | **3.00e+00 (1.84e+00)** | 1.08e+01 (3.02e+00) | 4.83e+00 (2.12e+00) | 4.83e+00 (1.43e+00) |
| Ackley-5D-2C-14% | **3.71e-02 (7.19e-04)** | 4.41e-02 (6.67e-03) | 7.94e-02 (1.46e-02) | 1.43e-01 (0.00e+00) | 1.11e-01 (2.45e-02) |
| Converter-36D-3C-27% | **9.50e-01 (1.77e-01)** | 1.32e+00 (2.31e-01) | 2.23e+00 (0.00e+00) | 2.23e+00 (0.00e+00) | 1.09e+00 (1.52e-01) |
| Vessel-4D-3C-78% | **2.06e-02 (1.39e-02)** | 1.20e+00 (3.90e-01) | 2.44e-01 (2.48e-01) | 3.79e+00 (1.25e+00) | 2.59e-02 (3.36e-02) |
| Car_Cabin-7D-8C-13% | 8.62e+00 (3.30e+00) | 1.40e+01 (2.14e+00) | 6.15e+00 (2.28e+00) | 5.75e+00 (2.03e+00) | **6.84e+00 (3.24e+00)** |
| Spring-3D-6C-0.38% | **6.40e+01 (2.83e+01)** | 1.11e+02 (1.50e+01) | 1.11e+02 (1.50e+01) | 1.11e+02 (1.50e+01) | 8.35e+01 (2.76e+01) |
| Experiment | COBAR-100 | CMES-IBO-100 | cEI-100 | SCBO-100 | SVM-CBO-100 |
| Rastrigin-1D-1C-60% | **2.21e+00 (1.41e+00)** | 2.84e+00 (1.62e+00) | 1.07e+01 (3.07e+00) | 4.83e+00 (2.12e+00) | 2.67e+00 (8.14e-01) |
| Ackley-5D-2C-14% | 3.69e-02 (3.08e-03) | **3.56e-02 (5.93e-03)** | 5.88e-02 (7.24e-03) | 1.43e-01 (0.00e+00) | 1.09e-01 (2.64e-02) |
| Converter-36D-3C-27% | **9.29e-01 (1.27e-01)** | 1.22e+00 (2.02e-01) | 2.14e+00 (1.75e-01) | 2.23e+00 (0.00e+00) | 9.73e-01 (1.45e-01) |
| Vessel-4D-3C-78% | **1.94e-02 (1.43e-02)** | 6.48e-01 (3.60e-01) | 1.51e-01 (1.25e-01) | 3.79e+00 (1.25e+00) | 2.23e-02 (8.96e-04) |
| Car_Cabin-7D-8C-13% | 6.40e+00 (2.72e+00) | 1.12e+01 (2.71e+00) | 5.92e+00 (2.34e+00) | 5.75e+00 (2.03e+00) | **6.03e+00 (2.40e+00)** |
| Spring-3D-6C-0.38% | **5.60e+01 (2.99e+01)** | 1.11e+02 (1.50e+01) | 1.11e+02 (1.50e+01) | 1.11e+02 (1.50e+01) | 8.35e+01 (2.76e+01) |

Table 3: Comparison of different methods' simple regrets across experiments. The table shows the updated experiment results after incorporating the SVM-CBO as an additional baseline. The upper block shows the simple regret at 70 iterations, while the lower shows the simple regret at 100 iterations. The standard error is shown in parentheses.

---

**Algorithm 2 COnstrained BO through Adaptive Region of Interest Acquisition (COBAR)**

1: **Input:** Search space $\mathbf{X}$, initial observation $\mathbf{S}_0$, horizon $T$, confidence factor $\delta$, confidence coefficient $\beta$;
2: **for** $t = 1\ to\ T$ **do**
3:     Update the posteriors of $\mathcal{GP}_{f,t}$ and $\mathcal{GP}_{\mathcal{C}_m,t}$
4:     Identify ROIs $\hat{\mathbf{X}}_t$, and undecided sets $U_{\mathcal{C}_m,t}$
5:     **for** $m \in \mathbf{M}$ **do**
6:       **if** $U_{\mathcal{C}_m,t} \neq \emptyset$ **then**
7:         Candidate for learning of each constraint:
        $\mathbf{x}_{\mathcal{C}_m,t} \leftarrow \arg\max_{\mathbf{x} \in \tilde{D}_{\hat{\mathbf{X}}_t} \cap U_{\mathcal{C}_m,t}} \alpha_{\mathcal{C}_m,t}(\mathbf{x})$ (4)
8:         $\mathcal{G} \leftarrow \mathcal{G} \cup \mathcal{C}_{m,t}$
9:     Candidate for optimizing the objective:
    $\mathbf{x}_{f,t} \leftarrow \arg\max_{\mathbf{x} \in \tilde{D}_{\hat{\mathbf{X}}_t}} \alpha_{f,t}(\mathbf{x})$ as in equation 3
10:     $\mathcal{G} \leftarrow \mathcal{G} \cup f$
11:     Maximize the acquisition from different aspects:
    $g_t \leftarrow \arg\max_{g \in \mathcal{G}} \alpha_{g,t}(\mathbf{x}_{g,t})$
12:     Pick the candidate to evaluate: $\mathbf{x}_t \leftarrow \mathbf{x}_{g_t,t}$
13:     Update the observation set
    $\mathbf{S}_t \leftarrow \mathbf{S}_{t-1} \cup \{(\mathbf{x}_t, y_{f,t}, \{y_{\mathcal{C}_m,t}\}_{m \in \mathbf{M}})\}$

---

### H.1 Additional Explanation of COBAR

For algorithm 2, $\{\mathbf{x}_{g_t,t}\}$ in line 11 are acquired in line 7 as $\mathbf{x}_{\mathcal{C}_m,t}$ or line 9 as $\mathbf{x}_{f,t}$, since $\mathcal{G}$ is composed of $\mathcal{C}_m$ and $f$. Roughly speaking, we are taking $\arg\max_{g,\mathbf{x}}$, yet we avoid using such notation for two reasons. (1) the domain where equation 5 and equation 6 are maximized are different; (2) the domain for equation 6 could even be empty. Therefore, we are currently taking the $\arg\max$ of equation 5 and equation 6 over different domains (if not empty) separately and then taking the $\arg\max$ of the corresponding acquisition function values as in line 11.

## I Discussions

Here, we offer additional explanation and discussion over COBAR.

### I.1 On the Comparability of Acquisition Functions over Different Underlying Functions

Both the acquisition functions for optimizing the objective and active learning are confidence interval-based, which reflects the uncertainty and is intrinsically comparable. With Assumption 1 that the black-box underlying functions are samples from the corresponding GPs specified by the kernels, we use the kernels to capture the scaling of the different unknowns. Our analysis does not assume that the kernels are the same, meaning that the theoretical results hold when the objective and constraints are of different scales. This analysis converts the algorithm's sensitivity to scale to the sensitivity

to hyper-parameter misspecification. In our experiments, we report the results when following the standard practice of kernel learning [Rasmussen and Williams, 2006] for both the proposed algorithm and baselines, as is stated at the end of the caption of figure 6. In summary, the compatibility is guaranteed by the properly specified kernel. Recent advancements in self-correcting BO [Hvarfner et al., 2024] or BO with unknown hyperparameters[Berkenkamp et al., 2019] propose various methods to address the challenge.

With regard to the practical concern over why the analysis does not require normalizing the different acquisition functions, the answer is threefold. First, since the correlation between the constraints and the objective is unknown, it is possible that the objective, in general, is of a smaller scale but bears the highest gradient near the boundary, meaning that the general scale of functions does not offer a guarantee to normalize the near-boundary uncertainties. Second, using the ROI to constrain the acquisition helps exclude the useless uncertainty reduction as the ROI considers both the objective and the constraints. If constraints dominate the COBAR acquisition, it suggests that the selected points remain likely to contain the global optimum as its objective does not have a high probability of being suboptimal. Such a query won't be wasted. A concrete example is illustrated in figure 3. Third, since we are assuming a universal upper bound for each constraint, the scale difference could make certain constraints dominant in the $\epsilon_m$. This could be addressed through the normalization of observations given prior knowledge.

## I.2   Difference from Other Existing CBO Methods with No-regret Guarantee

We briefly discuss the differences between COBAR and the previous theoretical results in CBO. Lu and Paulson [2022] addresses equality constraints for instantaneous penalty-based regret. However, the reward formulation is different. Lu and Paulson [2023] offers theoretical results on cumulative regret and violations. Yet, they assume querying points out of the feasible region still yields rewards and consider the violation separately.

In general, we are unaware that the existing CBO analysis results lead to a similar guarantee as in our work when assuming querying infeasible points does not yield a reward. One key difference is that with the active learning component and feasibility assumption, we could guarantee to query a feasible point that bears a reward converging to optimal value with the desired confidence. In our specific reward formulation, we regard such a guarantee and, therefore, the contribution in algorithm design and analysis as sufficiently different from the previous work, even when only focusing on the coupled setting.

## I.3   Empty Subsets of Search Space

It is possible that certain subsets discussed in section 3 could be empty at a certain $t$ as a result of intersections. However, according to the assumptions in section 3.4 and Lemma 1, the properly chosen $\beta$ does not result in over-aggressive filtering with high probability. From this perspective, ROI $\hat{\mathbf{X}}$ is soundly defined. COBAR is also robust to empty $U_{\mathcal{C}_m,t}$. As shown in algorithm 2, the domain where the acquisition functions defined in equation 4 and equation 3 are maximized allow empty $U_{\mathcal{C}_m,t}$ for COBAR to proceed.

## I.4   Limitations and Future Work

The limitation of COBAR includes (1) the inefficiency of identifying the ROIs due to the pointwise comparison in current implementation relying on discretization; (2) the lack of discussion over correlated unknowns, which are common in practice (e.g., two constraints are actually lower bound and upper bound of the same value). Though we briefly discuss and study corresponding scenarios, we expect the following work could improve the algorithm's effectiveness and the comprehensiveness of corresponding analysis accordingly.

