# OpenReview forum: "Finding Interior Optimum of Black-box Constrained Objective with Bayesian Optimization"
_NeurIPS.cc/2024/Workshop/BDU — NeurIPS BDU Workshop 2024 Poster_

### Official Review · Reviewer_JGsE · 2024-09-28
**Review of "Finding Interior Optimum of Black-box Constrained Objective with Bayesian Optimization"**

**Rating:** 8
**Confidence:** 3

**Review:**

**Summary:**
This paper presents a novel constrained Bayesian optimization (CBO) framework aimed at efficiently finding interior optimum by leveraging active learning for level-set estimation.

**Pros:**
- The paper provides rigorous theoretical analysis, including high-probability simple regret bounds.
- The experimental analysis is also relatively thorough, comparing the performance of the proposed COBAR algorithm against three baselines across six different CBO tasks -- two synthetic and four real-world -- with dimensions ranging from 1 to 32, number of constraints ranging from 1 to 8, and feasible region portion ranging from 0.38% to 78%.
- I like the additional illustrations in the appendix. They help understanding the proposed method. However, I would suggest increasing the font size in the plots to improve readability.

**Cons:**
- The paper briefly mentions that when max LCB is constant, the acquisition function is equivalent to UCB, but it does not explain why max LCB is used when there is a non-empty feasible region and why the regular LCB is used when there is no feasible region. Including such details can add clarity to the proposed COBAR algorithm.
- While the caption of Figure 1 explains the plot titles, there is a lack of clear correspondence between the shorthand task names used in the plots and the detailed descriptions in the experiment section (e.g., you can mention "Rastrigin and Ackley are synthetic", "Converter is the one extracted from UCI Machine Learning repository", etc.), particularly when the order of the tasks in the figure does not match the order in the experiment descriptions (e.g., Converter is mentioned last in the text but appears third in the plots). This inconsistency can confuse readers trying to map the tasks in the plots to the descriptions in the paper. Additionally, there appears to be a typo in one of the plot titles: 36D should be 32D, as indicated in the texts of the experiment section.

---

### Official Review · Reviewer_u3mL · 2024-10-08
**Solid theoretical foundation, but the experimental results require further explanation and commentary.**

**Rating:** 5
**Confidence:** 3

**Review:**

This paper presents COBAR, a novel algorithm for Constrained Bayesian Optimization (CBO) that effectively identifies the interior optimum of a black-box objective function under multiple black-box constraints. The primary contribution of the paper is the introduction of a region of interest (ROI) identification framework that refines the search space using Gaussian Process (GP) models, which combine active learning for level-set estimation to handle constraints and optimize the objective. The authors provide a solid theoretical foundation with thorough proofs of their theorems, although some unusual trends in the experiments warrant further explanation.

**Questions for the Authors**

- Please correct the typos (e.g., lower/upper case issues) in the reference section.

- The Bayesian Optimization (BO) framework relies heavily on Gaussian Processes (GPs) for modeling both the objective and the constraints. How does the method perform when the GP is a poor fit for the problem, and what kernel was chosen for the GP (e.g., squared exponential, Matérn, gamma-exponential)? Could the performance vary significantly with different kernel choices?

- The authors emphasize estimating the superlevel-set and focusing on it for optimization. How does the algorithm ensure sufficient exploration of the undecided set (i.e., regions where constraint satisfaction is uncertain), and could this lack of exploration affect the discovery of the global optimum?

- The combined ROI is formed by intersecting the ROIs of the constraints and the objective. In high-dimensional problems or when constraints are very tight, does this approach risk overly narrowing the search space and potentially missing promising areas?

- The absence of a pre-specified threshold for the objective function could introduce subjectivity in superlevel-set identification. Could this lead to overconfidence in certain regions of the search space, particularly if the confidence bounds are inaccurate?

- Regarding Figure 3, could the authors clarify why the labels for the "General ROI" and the "ROI for f/c" are the same across all five rows? This repetition makes the figure a bit confusing to interpret.

- In Figure 7, for the Ackley-5D-2C-14% task, the simple regret remains almost unchanged over iterations for $\beta=4$. This behavior appears to be unique to this specific case. Could the authors provide some insight into why this occurs?

- In Figure 9, I have a few specific questions that need clarification:

(1) Why does cEI perform poorly and show high regret in tasks like Converter, but performs better in others like Ackley and Vessel?
(2) Why does CONFIG show no improvement (flatline) in tasks like Rastrigin-1D and Ackley-5D?
(3) What causes COBAR’s performance to plateau in the Spring task? Is this behavior tied to the small feasible region size or other  factors?
(4) Why does SCBO exhibit high variance in tasks like Vessel and Spring? Could this be due to sensitivity to initialization or the structure of the problem?

---

### Decision · Program_Chairs · 2024-10-09

Accept (Poster)